# Structural basis of neurosteroid anesthetic action on GABA$_A$ receptors

Qiang Chen [1], Marta M. Wells[1,2], Palaniappa Arjunan[1,3], Tommy S. Tillman[1], Aina E. Cohen[4], Yan Xu [1,3,5,6] & Pei Tang[1,2,3]

Type A γ-aminobutyric acid receptors (GABA$_A$Rs) are inhibitory pentameric ligand-gated ion channels in the brain. Many anesthetics and neurosteroids act through binding to the GABA$_A$R transmembrane domain (TMD), but the structural basis of their actions is not well understood and no resting-state GABA$_A$R structure has been determined. Here, we report crystal structures of apo and the neurosteroid anesthetic alphaxalone-bound desensitized chimeric α1GABA$_A$R (ELIC-α1GABA$_A$R). The chimera retains the functional and pharmacological properties of GABA$_A$Rs, including potentiation, activation and desensitization by alphaxalone. The apo-state structure reveals an unconventional activation gate at the intracellular end of the pore. The desensitized structure illustrates molecular determinants for alphaxalone binding to an inter-subunit TMD site. These structures suggest a plausible signaling pathway from alphaxalone binding at the bottom of the TMD to the channel gate in the pore-lining TM2 through the TM1–TM2 linker. The study provides a framework to discover new GABA$_A$R modulators with therapeutic potential.

[1] Department of Anesthesiology and Perioperative Medicine, University of Pittsburgh, Pittsburgh, PA 15260, USA. [2] Department of Computational and Systems Biology, University of Pittsburgh, Pittsburgh, PA 15260, USA. [3] Department of Pharmacology and Chemical Biology, University of Pittsburgh, Pittsburgh, PA 15260, USA. [4] Stanford Synchrotron Radiation Lightsource, SLAC National Accelerator Laboratory, Menlo Park, CA 94025, USA. [5] Department of Structural Biology, University of Pittsburgh, Pittsburgh, PA 15260, USA. [6] Department of Physics and Astronomy, University of Pittsburgh, Pittsburgh, PA 15260, USA. Correspondence and requests for materials should be addressed to P.T. (email: ptang@pitt.edu)

Type A γ-aminobutyric acid receptors (GABA_ARs) control neuronal excitability and are the primary inhibitory pentameric ligand-gated ion channels (pLGICs) in the central nervous system[1]. Naturally occurring GABA_ARs are mostly heteropentamers assembled by homologous subunits. $(αβ)_2δ$GABA_ARs are exclusively extrasynaptic and $(αβ)_2γ$GABA_ARs are both synaptic and extrasynaptic[2] receptors that mediate prolonged tonic and short phasic inhibition, respectively[1,3]. Among them, the α1-containing receptors are the most abundant GABA_ARs in the brain[2].

GABA_AR-mediated inhibition of neuronal excitability results from membrane hyperpolarization due to $Cl^-$ flux upon GABA_AR activation, which can be triggered by binding of the neurotransmitter GABA to the orthosteric site in the extracellular domain (ECD) or allosteric binding of endogenous neuroactive steroids[1,3]. In addition, inhibitory functions of GABA_ARs can also be regulated by a wide variety of synthetic drugs in different physiological and pathological contexts. GABA_ARs are targets for the treatment of neurological diseases and disorders, such as epilepsy, depression and insomnia[3]. GABA_ARs are also targets for general anesthetics. The underlying mechanisms of action of these drugs on GABA_ARs remain to be determined[4].

Ample experimental evidence suggests that the transmembrane domain (TMD) of GABA_ARs harbors sites for the primary actions of general anesthetics and neurosteroids[5–12]. The TMD has an essential role in functional transitions among the resting, activated, and desensitized states of these $Cl^-$-conducting channels. Three-dimensional GABA_AR structures, particularly those revealing insights into how general anesthetics and neurosteroids exert their actions, are limited due to many technical challenges. For a long time, the crystal structure of a desensitized homomeric β3GABA_AR[13] was the only determined structure in the family of GABA_ARs. The successes in crystallographic structural determination of GLIC-α1GABA_AR[11] and β3-α5GABA_AR[12] chimeras open a new path to accelerate the process and demonstrate the feasibility of chimeric GABA_ARs in illustrating the structural basis underlying the actions of neurosteroids or general anesthetics. All of these structures represent the desensitized state. More recently, cryo-electron microscopy structures of the human α1β2γ2GABA_AR in complex with GABA and flumazenil have been published[14]. However, no apo-state structure has yet been determined for GABA_ARs.

Here, we report crystal structures of an α1GABA_AR chimera in the apo state and in an alphaxalone-bound desensitized state. Alphaxalone (5α-pregnan-3α-ol-11,20 dione) is a potent neurosteroid anesthetic. The anxiolytic, anticonvulsant, analgesic, and sedative-hypnotic effects of alphaxalone have been linked to its potentiation of GABA-evoked currents and direct activation of GABA_ARs[15,16]. However, the data about the alphaxalone binding site in GABA_ARs and the underlying structural basis of alphaxalone action are sparse[17,18]. Our crystal structures illustrate the molecular details of alphaxalone binding to the α1GABA_AR TMD, reveal neurosteroid anesthetic action starting at the bottom of the TMD, and provide a structural basis for the rational discovery of new drugs.

## Results

### The α1GABA_AR chimera resembles functions of GABA_ARs.
We constructed a GABA_AR chimera by fusing the TMD of human α1GABA_ARs to the ECD of ELIC, a prokaryotic pLGIC from *Erwinia chrysanthemi*[19] (Fig. 1a). The final residue R199 in the ECD of ELIC was fused with the first residue K222 in the TMD of human α1GABA_AR. To facilitate crystallization, the large intracellular loop between TM3 and TM4 in α1GABA_AR (G313 to N387) was replaced by the tripeptide linker from ELIC (GVE)

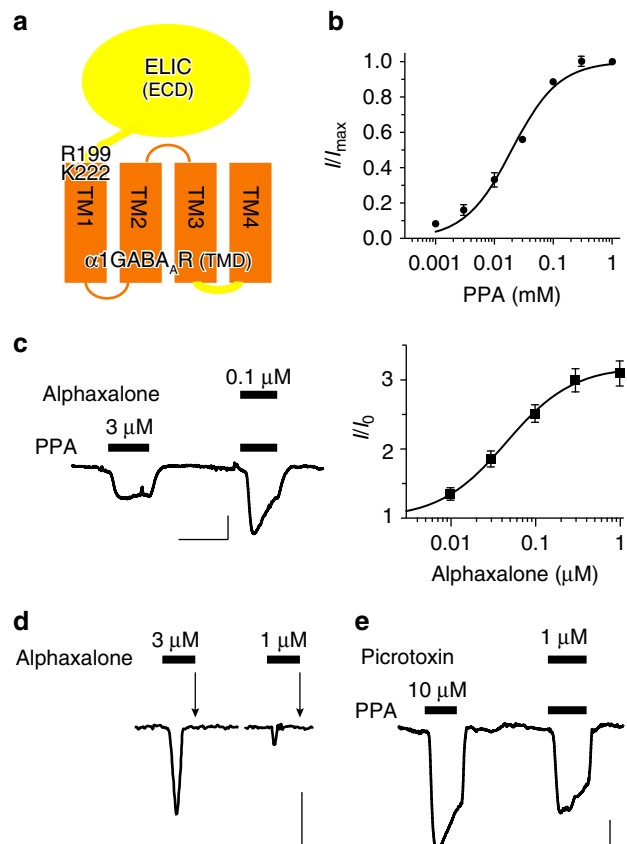

**Fig. 1** Construction and function of the ELIC-α1GABA_AR chimera.
**a** Schematic representation of the ELIC-α1GABA_AR chimera, constructed by fusing the ELIC extracellular domain (yellow) ending at residue R199 to the α1GABA_AR transmembrane domain (orange) beginning at residue K222. See more details in the supporting information (Supplementary Fig. 1). **b** The ELIC agonist propylamine (PPA) activates the α1GABA_AR chimera expressed in *Xenopus* oocytes in a concentration dependent manner ($EC_{50} = 19.8 ± 1.5 μM$, $n = 4$). **c** The neurosteroid alphaxalone potentiates the current of oocytes expressing the α1GABA_AR chimera: (left) representative potentiation trace by 0.1 μM alphaxalone; (right) concentration-response potentiation curve with $EC_{50} = 45.5 ± 10.7 nM$ ($n = 6$). **d** Representative traces showing that 3 or 1 μM alphaxalone activates and then quickly desensitizes the α1GABA_AR chimera (current disappears during alphaxalone application as marked by arrows). **e** Picrotoxin, a known GABA_AR blocker, inhibits the α1GABA_AR chimera. Note the quick desensitization by 10 μM PPA. Error bars in **b** and **c** represent SEM. Scale bars in **c**, **d**, and **e** represent 30 s (horizontal) and 10 nA (vertical)

and 12 residues at the C-terminus of α1GABA_AR (R418 to Q429) were removed. Sequence details of the chimera are provided in the supplementary information (Supplementary Fig. 1). Probably because the human α1GABA_AR and ELIC share 46.6% sequence similarity in their ECDs and because GABA also activates ELIC[20], the α1GABA_AR chimera is functional in *Xenopus* oocytes without sequence optimization at the ECD-TMD interface. This easy process of obtaining a functional α1GABA_AR chimera is in contrast with difficulties in making the ELIC-α7 nicotinic acetylcholine receptor (nAChR) chimera, which required extensive sequence modification at the ECD-TMD interface in order to mimic the functions of α7nAChR[21]. ELIC agonists[20], including propylamine (PPA), activate oocytes expressing the α1GABA_AR chimera in a concentration dependent manner (Fig. 1b). The neurosteroid anesthetic alphaxalone potentiates the agonist-induced current (Fig. 1c), directly activates the α1GABA_AR

chimera in the absence of agonist and desensitizes the channel during its continued application (Fig. 1d). Picrotoxin, known to inhibit GABA$_A$Rs, also inhibits the α1GABA$_A$R chimera (Fig. 1e). These functional responses of the α1GABA$_A$R chimera to alphaxalone and picrotoxin resemble those observed on the authentic α1-containing GABA$_A$Rs[22–24]. Alphaxalone inhibits ELIC (Supplementary Fig. 2). The distinctly different functional and pharmacological properties of ELIC and GABA$_A$Rs[25,26] offer an opportunity to dissect the role of the GABA$_A$R TMD in a chimeric form.

**X-ray structures of apo and desensitized α1GABA$_A$R chimeras.** The α1GABA$_A$R chimera was expressed in *E. coli* and purified in DDM micelles, in which the chimera forms homogenous pentamers suitable for crystallization (Supplementary Fig. 3). We determined x-ray structures of the α1GABA$_A$R chimera in the apo and desensitized states with 3.2 and 3.45 Å resolutions, respectively (Table 1; Fig. 2). Alphaxalone activates and subsequently desensitizes the α1GABA$_A$R chimera within a minute. Thus, on the crystallization time scale, the alphaxalone-bound α1GABA$_A$R chimera is in a desensitized state (Fig. 1d).

The desensitized (Fig. 2a) and apo (Fig. 2b) α1GABA$_A$R chimeras retain the common structural characteristics of

**Table 1 Crystallographic data collection and refinement statistics**

| | Alphaxalone-bound ELIC-α1GABA$_A$R (PDB: 6CDU)[a] | Apo ELIC-α1GABA$_A$R (PDB: 6D1S)[a] |
|---|---|---|
| *Data collection* | | |
| Space group | P2$_1$ | P2$_1$ |
| *Cell dimensions* | | |
| *a, b, c* (Å) | 108.2, 263.5, 109.2 | 108.5, 264.8, 109.2 |
| *α, β, γ* (°) | 90.0, 110.9, 90.0 | 90.0, 110.5, 90.0 |
| Wavelength (Å) | 0.9762 | 0.9756 |
| Resolution (Å) | 40.00–3.45 (3.52–3.45) | 40.00–3.20 (3.25–3.20) |
| $R_{merge}$ | 0.156 (3.252) | 0.104 (2.470) |
| $R_{pim}$ | 0.042 (0.877) | 0.041 (0.961) |
| $<I/\sigma>$ | 13.2 (1.1) | 11.4 (1.0) |
| CC$_{1/2}$ | 0.999 (0.488) | 0.996 (0.345) |
| Completeness (%) | 97.8 (99.3) | 99.0 (98.9) |
| Redundancy | 14.4 (14.4) | 7.3 (7.5) |
| *Refinement* | | |
| Resolution (Å) | 39.88–3.45 | 37.77–3.20 |
| No. reflections | 73,029 | 93,532 |
| $R_{work}/R_{free}$ | 0.227/0.286 | 0.220/0.269 |
| *No. atoms* | | |
| Protein | 25,690 | 25,690 |
| Alphaxalone | 240 | |
| *B-factors (Å$^2$)* | | |
| Protein | 142.0 | 142.9 |
| Alphaxalone | 134.3 | – |
| *R.m.s deviations* | | |
| Bond lengths (Å) | 0.003 | 0.003 |
| Bond angles (°) | 0.661 | 0.703 |
| MolProbity score | 1.90 | 1.96 |
| *Ramachanran* | | |
| Favored (%) | 93.01 | 92.63 |
| Allowed (%) | 6.41 | 6.95 |
| Outliers (%) | 0.58 | 0.42 |
| Rotamer outliers (%) | 0 | 0 |

[a]Merged from two datasets

pLGICs[11–13,19,27–34], including well-folded β-sheets in the ECD and α-helices in the TMD. The pentameric assembly provides a central pore for ion permeation (Fig. 2a). In addition to the peptide bond linking the last ECD residue R199 of ELIC to the first TMD residue K222 of α1GABA$_A$R (Fig. 1a), the ECD and TMD in the chimera are structurally coupled by extensive interface interactions (Fig. 2b) that have been suggested to be crucial for functional channels[35,36]. Notably, K279 in the TM2–TM3 loop, a conserved residue in GABA$_A$Rs, engages in a polar interaction with T28 of the β1–β2 linker in the adjacent subunit. A *cis*-conformation of P120, a highly conserved residue in all pLGICs, leads to a Cys-loop orientation that enables the backbone carbonyl of neighboring residues (F119 and F116) to participate in polar contacts with the TM3 residue A285 and Y282 in the TM2–TM3 loop. These ECD–TMD interactions have also been reported for structures of β3GABA$_A$R[13] and GLIC-α1GABA$_A$R[11].

The apo α1GABA$_A$R chimera shows little spontaneous leaking current in electrophysiology measurements and its x-ray structure offers a glance at the transmembrane pore of α1GABA$_A$R in the apo state (Fig. 3). The pore-lining TM2 helix in each subunit is oriented with 10.2° radial and 2.3° lateral tilting angles so that the pore radius is largest at the extracelluar end and is reduced gradually toward the intracellular end. The most constricted radius (~2.0 Å) is at V257 (2′) (Fig. 3b, c). The pore radii at both V257 (2′) and P253 (−2′) are too small to allow passage of a hydrated Cl$^-$ ion. This closed channel shows no contraction at L264 (9′) because the L264 sidechains are tangential to the pore axis. The pore profile is considerably different from those reported previously for resting-state pLGICs (Fig. 3c, Supplementary Fig. 4), including ELIC[19,37], GLIC[38], GluCl[29], and 5HT$_{3A}$R[33], for which the conserved leucine (9′) constitutes the narrowest pore or channel gate. However, a previous study of α1β1γ2GABA$_A$R using the scanning-cysteine-accessibility method (SCAM) reported that, in the absence of GABA, charged sulfhydryl reagents applied from the extracellular end of the resting-state channel were able to penetrate to the level of α1V257 (2′)[39], suggesting a similar pore profile to that shown in the structure of the apo α1GABA$_A$R chimera (Fig. 3).

The pore profile of the α1GABA$_A$R chimera desensitized by alphaxalone is similar to the pore profile of the apo chimera, except that the most constricted pore is located at P253 (−2′) in the desensitized structure (Fig. 3c). The putative gate at the −2′ position in the desensitized α1GABA$_A$R chimera matches with observations from the structures of desensitized GLIC-α1GABA$_A$R[11], β3-α5GABA$_A$R[12], β3GABA$_A$R[13], α3[40], and α1[34] glycine receptors (GlyRs), and α4β2nAChR[27] (Fig. 3c, Supplementary Fig. 4).

Global twisting and blooming movements of pLGICs have been proposed to accompany functional conformation changes[29,38,41]. Both the apo and alphaxalone-bound ELIC-α1GABA$_A$R are closed channels (Fig. 3) and no significant blooming is observed. Relative to the aligned structure of the apo ELIC-α1GABA$_A$R, the alphaxalone-bound structure shows 1.3° inward and 0.18° outward radial tilt from the pore axes in the ECD and TMD, respectively. The alphaxalone-bound structure shows a small counterclockwise twist (0.80°) of the ECD and a clockwise twist (1.37°) of the TMD around the pore axis (Supplementary Fig. 5). This direction of twist leads toward channel opening based on a structure survey of pLGICs under different functional states[29,34,38]. In the case of ELIC-α1GABA$_A$R, however, the small magnitude of twist in the desensitized alphaxalone-bound structure with respect to the apo structure is consistent with the fact that both channels are closed (Fig. 3).

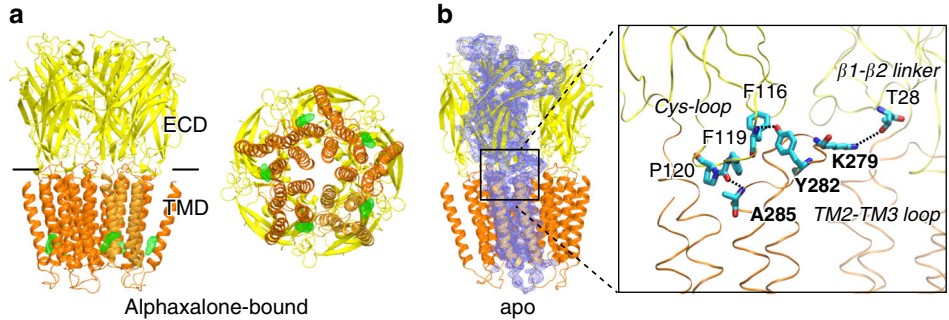

**Fig. 2** Crystal structures of the ELIC-α1GABA_AR chimera. **a** Side (left) and bottom (right) views of the α1GABA_AR chimera in complex with alphaxalone. Alphaxalone binding to the inter-subunit sites in the TMD is indicated by the $F_O$–$F_C$ omit electron density map contoured at 4 $\sigma$ (green mesh). **b** Side view of the apo α1GABA_AR chimera. One of the five subunits is covered with the 2$F_O$–$F_C$ electron density map contoured at 1 $\sigma$ (blue mesh). A zoom-in view of the interfacial region shows the representative residue contacts at the interface between the TM2–TM3 loop and the Cys-loop or β1–β2 linker in the ECD: F119-A285 (3.1 Å), F116-Y282 (2.4 Å), T28-K279 (4.5 Å)

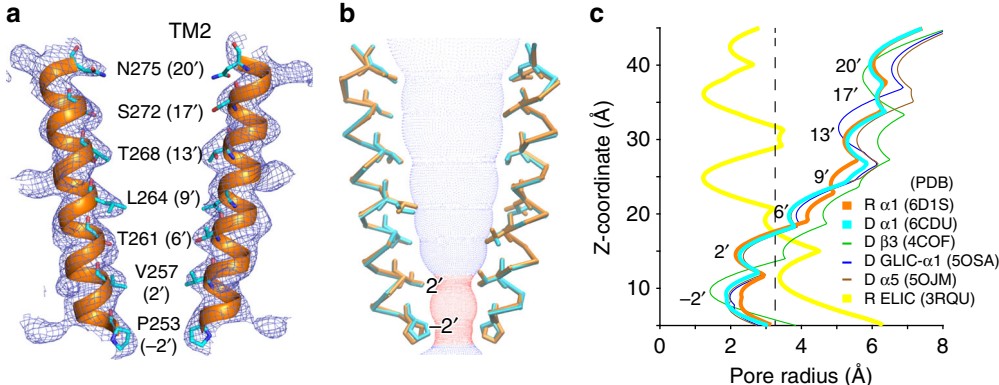

**Fig. 3** Crystal structures of the pore of the ELIC-α1GABA_AR chimera. **a** Pore lining residues in the TM2 helices of the apo α1GABA_AR chimera covered by the 2$F_O$–$F_C$ electron density map contoured at 1 $\sigma$ (blue mesh). **b** Overlaid structures of the pore-lining TM2 helices from apo (orange) and alphaxalone-bound (cyan) α1GABA_AR chimeras. Blue and red dots define apo α1GABA_AR chimera pore radii greater or less than the radius of a hydrated Cl⁻ ion (3.2 Å), respectively. **c** Comparison of the pore radii of the α1GABA_AR chimera in the apo (orange) and desensitized states (cyan) to the desensitized β3GABA_AR (green), the desensitized GLIC-α1GABA_AR chimera (blue), the desensitized α5GABA_AR chimera (brown), and resting ELIC (yellow)

**The binding mode of alphaxalone.** The structure of the α1GABA_AR chimera co-crystallized with alphaxalone shows that alphaxalone binds to five equivalent inter-subunit sites close to the intracellular end of the α1GABA_AR TMD (Fig. 2a, Supplementary Fig. 6a). Residues within 4-Å of alphaxalone are from TM3 of the principal subunit (A305, T306, Y309, and F310) and from TM1 (Q242, V243, W246) and TM4 (P401) of the neighboring complementary subunit (Fig. 4, Supplementary Fig. 6b). The aromatic ring of the conserved residue W246 is parallel to the C ring of alphaxalone, establishing a ring stacking interaction that potentially stabilizes the binding. The polar sidechains of Q242 and T306 are within reach to form putative hydrogen bonds with alphaxalone (Fig. 4b).

To assess the importance of these interactions to the functional modulation of α1GABA_AR by alphaxalone, we performed electrophysiology measurements on the wild type (WT) and three mutants of the α1GABA_AR chimera. Two of these mutants, Q242L and T306A, are similar to the WT α1GABA_AR chimera in their response to the orthosteric agonist PPA, whereas the PPA EC_50 of the W246L mutant is increased two orders of magnitude compared to WT (Fig. 4c), signaling the importance of this conserved residue W246 in the function of GABA_ARs. For the Q242L and T306A mutants, 0.1 μM alphaxalone still potentiates channel currents but with smaller magnitudes compared to the WT chimera (Fig. 4d). The W246L mutation

has a stronger effect and almost completely abolishes potentiation by the same concentration of alphaxalone (Fig. 4d). As expected, these mutations also reduce channel activation by alphaxalone (Fig. 4e). These findings from the α1GABA_AR chimera are consistent with the results from the full-length GABA_ARs. A number of previous studies on full-length GABA_ARs show that the α1-W246L and α1-Q242L mutations significantly suppressed allopregnanolone potentiation[42,43]. Similarly, alphaxalone potentiation of α1β3 GABA_AR is also significantly reduced in the α1-T306A mutant compared to the WT receptor (Supplementary Fig. 7).

We performed molecular dynamics (MD) simulations (Fig. 5, Supplementary Fig. 8) to quantify the stability of alphaxalone interactions with W246, Q242, and T306. The stability of ring stacking is measured by the distances between two pairs of atoms in alphaxalone and W246 (Fig. 5a, b). The narrow distance distributions over the course of simulations suggest stable ring stacking. The same simulation data are used to estimate probabilities of forming hydrogen bonds between relevant hydroxyl and carbonyl groups in alphaxalone and Q242 or T306 (Fig. 5c), showing that alphaxalone has a ~ 55% probability to form a hydrogen bond with Q242, but only a ~23% probability with T306. The data agree with mutagenesis and functional results (Fig. 4) that show a stronger influence of Q242 than T306 on functional effects of alphaxalone.

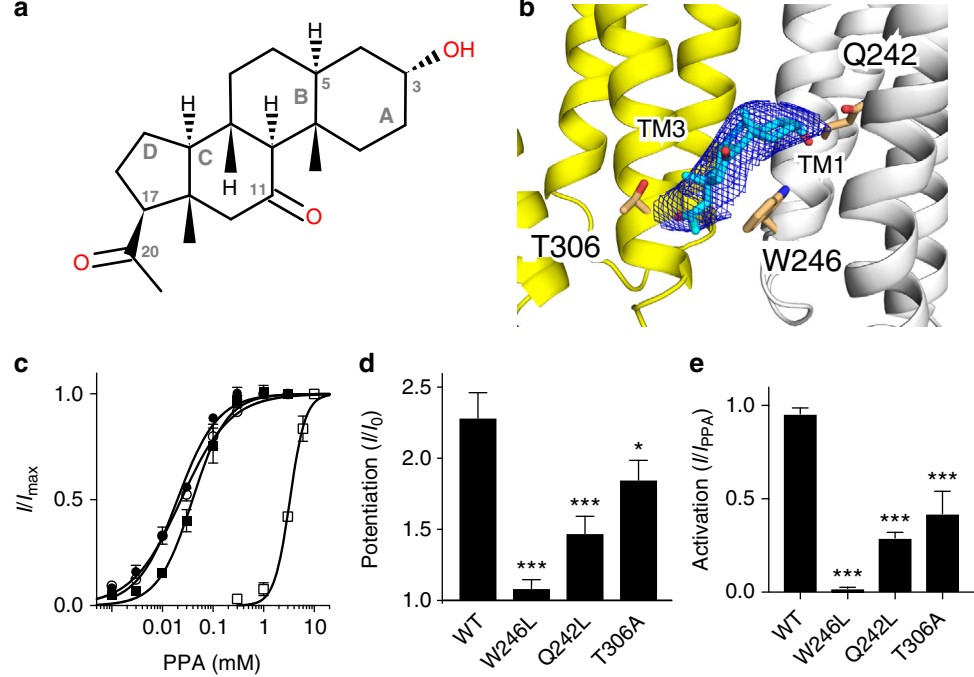

**Fig. 4** Alphaxalone binding mode in the α1GABA$_A$R chimera. **a** 2D chemical structure of alphaxalone with rings and carbon atoms labeled according to the IUPAC standard for steroids. **b** Crystal structure of alphaxalone (cyan) bound to a pocket lined by residues in the transmembrane domain from the principal (yellow) and complementary (white) subunits of the α1GABA$_A$R chimera. Alphaxalone is surrounded by the $2F_O$–$F_C$ electron density map contoured at 1 $\sigma$ (blue mesh). Three residues in close contact with alphaxalone are highlighted. Functional validation of the alphaxalone-binding site was performed by **c** activation of *Xenopus* oocytes expressing WT (solid circles), T306A (open circles), Q242L (solid squares) and W246L (open squares) α1GABA$_A$R chimeras by propylamine (PPA) with EC$_{50}$ = 20 ± 1, 23 ± 2, 39 ± 3, and 3300 ± 300 μM, respectively; **d** alphaxalone (0.1 μM) potentiation at the EC$_{10}$ concentration of PPA; **e** alphaxalone (3 μM) activation normalized to EC$_{100}$ PPA activation for each construct. Error bars represent SEM ($n \geq 3$ oocytes). Statistical significance was assessed by one-way ANOVA followed by Fisher's LSD post-hoc test. Asterisks indicate statistical difference from WT at $p < 0.001$ (***) and $p < 0.05$ (*)

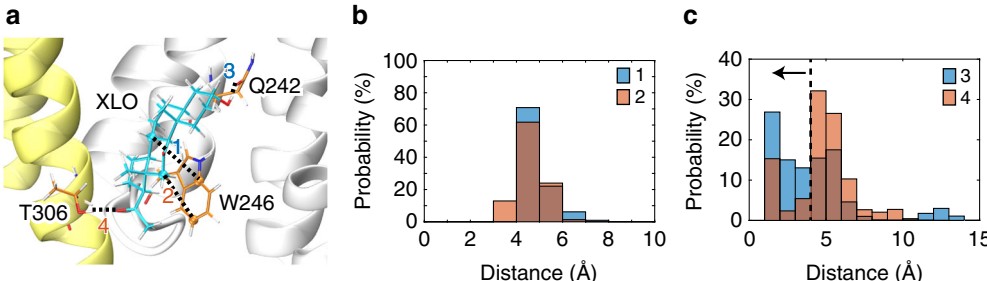

**Fig. 5** Alphaxalone interactions with nearby residues in molecular dynamics (MD) simulations. **a** A representative snapshot from MD simulations shows alphaxalone contacts with T306, Q242, and W246. Distances between alphaxalone and the residues are measured as marked by the dash lines. **b** Histograms of distances between alphaxalone and W246 atoms (1 and 2) shown in **a**. **c** Histograms of distances between alphaxalone and Q242 or T306 atoms (3 and 4, respectively) shown in **a**. Data in **b** and **c** are from three replicate 50-ns simulations, where snapshots were collected every 100-ps for analysis. Distances were measured for each of the five alphaxalone molecules per replicate simulation (500 snapshots × 3 replicates × 5 alphaxalone = 7500 distances in total)

**A path for alphaxalone activation and desensitization.** In the absence of agonist binding to the orthosteric site in the ECD, alphaxalone allosterically triggers the activation and subsequent desensitization of ELIC-α1GABA$_A$R (Fig. 1d). What is the structural basis for channel activation and desensitization by alphaxalone? In addition to the small twist movement of the ECD and TMD along the channel axis (Supplementary Fig. 5), we notice a set of conformational changes in the aligned x-ray structures of the apo and alphaxalone-bound ELIC-α1GABA$_A$R (Fig. 6, Supplementary Fig. 9). Alphaxalone binding to the bottom of the TMD introduces structural changes, including the orientation of W246 in the TM1, presumably to optimize its

interaction with alphaxalone (Fig. 6). The change passes to the TM1–TM2 linker and further to the TM2, particularly at P253 (−2′) and V257 (2′), resulting in a backbone RMSD of 0.95 Å for the residues covering from W246 in TM1 to V257 (2′) in TM2 (Fig. 6b). The structural changes further propagate up to the ECD–TMD interface and other ECD regions, as evidenced by the observed RMSDs: 0.66 Å for the TM2–TM3 loop (P278-T284), 0.61 Å for the pre-TM1 (R199-I223), 0.58 Å for the β1–β2 linker (V26-E30), 0.63 Å for the Cys-loop (N112-F126), 1.01 Å for loop A (N80-S84), and 0.84 Å for loop C (D172-N186) (Supplementary Fig. 9). Although these small changes do not necessarily reflect the actual magnitude of the structural changes involved in

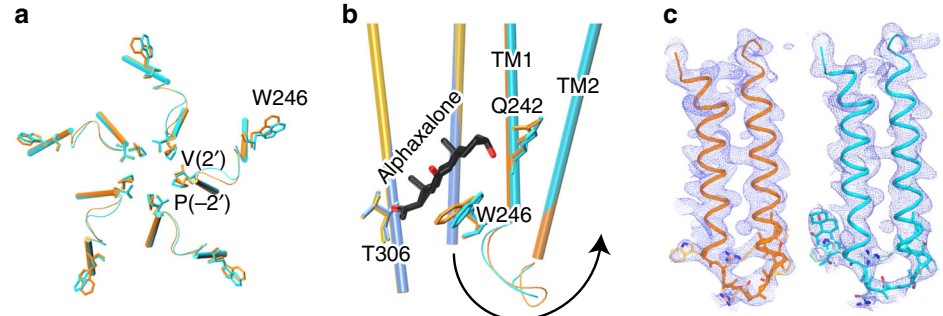

**Fig. 6** Alphaxalone-induced structural changes at the bottom of the TMD. **a** Bottom view of overlaid TM1-TM2 structures of the apo (orange) and alphaxalone-bound (cyan) ELIC-α1GABA$_A$R. **b** Side view of overlaid structures of apo (principal subunit - gold; complementary subunit - orange) and alphaxalone-bound (principal subunit - blue; complementary subunit - cyan) ELIC-α1GABA$_A$R. For clarity, only TM2 and TM3 are shown in the principal subunit and only TM1 and TM2 are shown in the complementary subunit. The arrow highlights structural perturbations originating from the alphaxalone binding site near W246 through the TM1–TM2 linker to the pore-lining residues P253 (−2′) and V257 (2′). **c** The 2$F_O$-$F_C$ electron density maps (blue mesh, contoured at 1 $\sigma$) covering TM1–TM2 in the apo (left) and alphaxalone-bound (right) ELIC-α1GABA$_A$R. The sidechains for residues W246 to V257 (2′) are highlighted

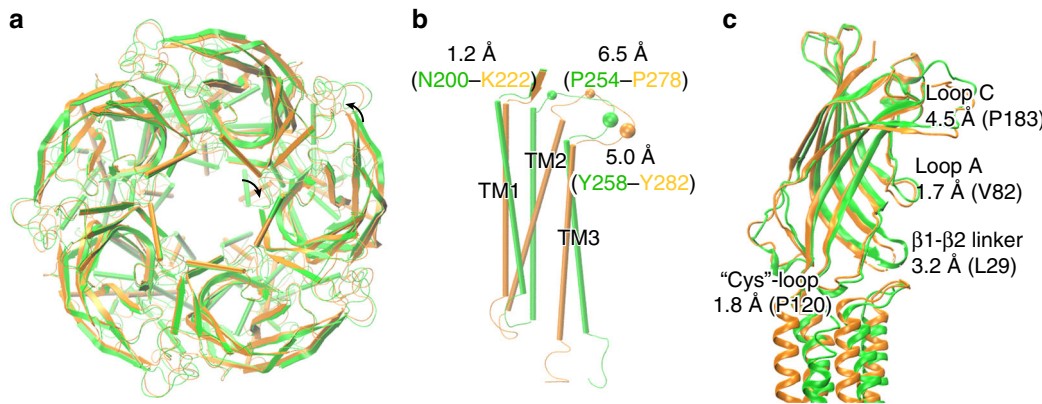

**Fig. 7** Structural changes in the ECD due to different TMDs. **a** Top view of crystal structures of apo ELIC-α1GABA$_A$R (orange) and apo ELIC (green), aligned along all ECD residues. **b** Displacements of Cα atoms of equivalent residues in apo ELIC-α1GABA$_A$R and apo ELIC in the pre-TM1 and the TM2–TM3 loop are labeled. Both the pre-TM1 and TM2–TM3 loop regions may affect the ECD. **c** Structural changes in the ECD are measured by displacements of the highlighted residues in several key regions

the functional transitions from the resting to activated and subsequent desensitized states, they suggest a plausible path starting at the bottom of the TMD for alphaxalone-induced channel activation and desensitization.

**Structural cooperation and independence of the ECD and TMD.** The structural independence of individual domains in pLGICs and cooperativity between these domains are essential for constructing functional chimeric channels[11,12,44]. In this study, ELIC-α1GABA$_A$R provides a window into how the ECD and TMD both retain their individuality in the chimera and cooperate structurally to form a functional channel. Crystal structures of apo ELIC-α1GABA$_A$R and apo ELIC (aligned along their common ECDs) show that even under the same ECD, the chimera TMD independently adopts the α1GABA$_A$R structure that is significantly different from the ELIC TMD structure (Fig. 7a, b). The ECD influence on the TMD structure is rather limited. Indeed, even an isolated TMD of pLGICs can form functional channels that retain some characteristics of the parent channels[45–47]. The structural cooperativity between the individual domains is also observed (Fig. 7). Although most parts of the ECD in the two structures overlap well, there is an overall ECD backbone RMSD of 2.1 Å, resulting mainly from the displacement of several key regions in the chimera (Fig. 7c). The subtle and

profound displacements of the respective pre-TM1 and the TM2–TM3 loop in the chimera (Fig. 7b) may have led to the conformational adjustments of the Cys-loop and β1–β2 linker at the ECD–TMD interface, as well as alterations further up at loop A and loop C (Fig. 7c). As an example of structural cooperation between the ECD and TMD, the conserved residue K279 in the TM2–TM3 of GABA$_A$Rs and its equivalent residue R255 in ELIC are oriented in different directions so that they form respective inter-subunit and intra-subunit polar interactions with T28 in the β1–β2 linker (Supplementary Fig. 10). The inter-subunit K279–T28 interactions would not be possible without the β1–β2 linker relocation in the chimera. An equivalent inter-subunit polar interaction (K274-E52) is also observed in the crystal structure of β3GABA$_A$R[13]. The importance of this lysine residue in channel gating of GABA$_A$Rs has been well recognized[35,36]. Mutation of this lysine impairs function of GABA$_A$Rs[36] and is associated with epilepsy[48,49].

**Discussion**

We have determined both the apo and desensitized structures for the ELIC-α1GABA$_A$R chimera. Although our desensitized structure was obtained through alphaxalone binding that directly activates and subsequently desensitizes the chimera (Fig. 1), it demonstrates the same gate in the pore at the cytosolic portal as

observed in the structures of β3GABA$_A$R, α1GlyR, α3GlyR, and α4β2nAChR[13,27,34,40] desensitized by orthosteric ligands (Supplementary Fig. 4) as well as the more recently determined GLIC-α1GABA$_A$R and β3-α5GABA$_A$R[11,12] (Fig. 3c). The consistency of the pore constriction at the −2′ position among these desensitized pLGICs, independent of whether they are desensitized by orthosteric (in most cases) or allosteric (such as in this study) ligands, settles the debate about what causes non-conductivity in a desensitized pLGIC[50–54].

No structure has been reported in the past for a resting GABA$_A$R. The apo ELIC-α1GABA$_A$R structure presents a look at the TMD of α1GABA$_A$R in the resting state. The pore profile shows the most constriction at V257 (2′) (Fig. 3), not at the conserved L264 (9′) that is perceived as a gate of channel activation based on previously published structures of the resting pLGICs, including GluCl[29], 5HT$_{3A}$R[31,33], GLIC[38], and ELIC[19,37] (Fig. 3, Supplementary Fig. 4). Because of the unconventional occlusion location in the pore of the apo ELIC-α1GABA$_A$R, it is natural to question whether the apo structure reflects a true resting-state conformation of ELIC-α1GABA$_A$R. We have analyzed the question from several different angles. Could the DDM detergent and crystallization conditions have introduced conformational biases to the resting ELIC-α1GABA$_A$R? DDM has been used for purification and crystallization of many different pLGICs, including GluCl that exhibits different pore conformations for the closed/resting state and open/activated state channels[29]. Both ELIC and ELIC-α1GABA$_A$R are crystallized with DDM, but show distinctly different pore conformations in the resting-state condition. Thus, it is not convincing to simply attribute the unconventional resting-state pore profile to detergent- and/or crystallization-related artifacts, even though one cannot completely rule out this possibility. Can we assign the apo ELIC-α1GABA$_A$R to a desensitized state in the absence of agonist because of the similar structures of the apo and desensitized ELIC-α1GABA$_A$R? The classic definition of desensitization requires the presence of agonists or channel modulators. Moreover, our electrophysiology functional data also do not support this assignment. Can a resting-state GABA$_A$R adopt the non-conducting channel conformation similar to those observed for desensitized pLGICs? Our apo structure suggests such a possibility. In addition, this possibility has been suggested previously by a SCAM study[39], in which the charged sulfhydryl reagents were able to penetrate from the ECD of α1β1γ2GABA$_A$R to the pore at the level of α1V257 (2′) in the resting state. We also notice that GABA$_A$Rs have a different pattern of hydrophobicity in pore-lining residues compared to those pLGICs with known resting structures (Supplementary Fig. 11, Supplementary Table 1), including the anion-conducting GluCl[29] along with the cation-conducting 5HT$_{3A}$R[31,33], *Torpedo* nAChR[55], GLIC[38], and ELIC[19,37], which have consecutive hydrophobic rings at the 9′, 13′ and 17′ (16′) positions. In contrast, GABA$_A$Rs have mostly hydrophilic pore-lining residues except at the 9′, −2′, and 2′ positions (Supplementary Fig. 11). Combining all aforementioned information with the pore radius profiles for the apo and desensitized α1GABA$_A$R chimera, we propose a channel gate for both activation and desensitization at the cytoplasmic end of the pore formed by the 2′ and −2′ residues. Whether the finding can be generalized to other GABA$_A$Rs mandates additional structural investigations, particularly with apo GABA$_A$Rs.

The binding site identified for the anesthetic alphaxalone overlaps well with the site elegantly illustrated previously for the endogenous potentiating neurosteroids tetrahydro-deoxycorticosterone (THDOC) and pregnanolone in the respective GLIC-α1GABA$_A$R and β3-α5GABA$_A$R chimeras[11,12] (Supplementary Fig. 12), suggesting that neurosteroid potentiation and activation of GABA$_A$Rs are mediated through this conserved

inter-subunit site close to the base of the TMD. It is also noteworthy that photoreactive analogs of the intravenous general anesthetics propofol[9] and etomidate[56,57] have been photolabeled to the TM1 residues M236 and I239, showing the proximity of the binding sites for these intravenous anesthetics and alphaxalone (Supplementary Fig. 6). A more recent mutagenesis/electrophysiology study of α1β3γ2$_L$GABA$_A$Rs also suggest the proximity between the alphaxalone site and the sites for propofol and etomidate at the transmembrane β$^+$-α$^-$ interface[18]. Among all the interactions between alphaxalone and α1GABA$_A$R shown in our structure, the ring stacking of the conserved W246 in the TM1 with the C ring of alphaxalone is most critical. Eliminating this interaction in the W246L mutant completely abolishes potentiation and activation by alphaxalone (Fig. 4). Putative hydrogen bonding, especially through Q242 (one helical turn above W246) to the 3α-hydroxyl of alphaxalone, is also important as the potentiation by alphaxalone was reduced substantially in Q242L and only moderately in T306A (Fig. 4). The order of W246, Q242, and T306 in their impact to functional modulation of alphaxalone is associated with the stability of their interactions with alphaxalone as revealed in MD simulations (Fig. 5), in which the ring stacking with W246 is arguably the steadiest interaction. The probability for alphaxalone to form a hydrogen bond with Q242 is much higher than with T306 (Fig. 5). A more dominant role of the complementary α1 subunit TM1 containing W246 and Q242 in alphaxalone potentiation agrees with the finding in heteromeric GABA$_A$Rs[42], where allopregnanolone binding to the β$^+$-α$^-$ interface is a major contributor to functional potentiation.

Neurosteroids and general anesthetics, including alphaxalone, not only potentiate but also directly activate GABA$_A$Rs at higher concentrations[10,15,16,58,59]. The structures of the α1GABA$_A$R chimera determined in this study suggest a mechanism suitable for direct activation by alphaxalone, in which the TM1–TM2 linker at the bottom of the TMD plays a crucial role in transducing conformational changes that originate from alphaxalone binding at the TM1 (W246 and Q242) to the pore-lining TM2 helices (Fig. 6). Because the same neurosteroid binding site in the TMD is also responsible for potentiation[11], the signaling pathway starting at the bottom of the TMD revealed in our crystal structures likely is relevant for both potentiation and activation of GABA$_A$Rs. Similar conformational changes are also found in neurosteroid potentiation of the β3-α5GABA$_A$R chimera, though the changes induced by neurosteroids in that case may be diluted by the channel's high propensity for spontaneous opening[12]. Our structures also show that conformational changes introduced by binding of the neurosteroid anesthetic alphaxalone and mediated by the TM1–TM2 linker can propagate beyond the TM2 (Supplementary Figs. 5, 9). Together, these results underscore three conclusions. First, alphaxalone binding to the inter-subunit site at the bottom of the TMD can introduce global conformational changes involved in channel potentiation, activation, and desensitization. Second, the TM1–TM2 linker, noted previously for its involvement in channel desensitization[60], plays a key role in mediating activation and potentiation by neurosteroids. Finally, this mechanism of activation or potentiation starting at the bottom of the TMD can be exploited in the rational search for new GABA$_A$R modulators with better potency and efficacy.

## Methods

**Protein expression and purification**. The ELIC-α1GABA$_A$R chimera was constructed using overlapping PCR by fusing the ELIC ECD ending at R199 with the human α1GABA$_A$R TMD starting at K222 (Supplementary Fig. 1). Primer sequences are provided in Supplementary Table 2. To facilitate crystallization, we replaced the lengthy intracellular loop (G314 to N387) of α1GABA$_A$R with the short linker (G290-V291-E292) connecting TM3 and TM4 in ELIC, and also deleted 12 residues at the C terminus of α1GABA$_A$R (REPQLKAPTPHQ). ELIC-α1GABA$_A$R, similar to the ELIC construct designed for *E. coli* expression[37], was

cloned in pET26b vector under an IPTG-inducible promoter. N-terminally His-tagged maltose binding protein (MBP) was fused to the N terminus of ELIC-α1GABA$_A$R and a TEV enzyme cleavage site was inserted between MBP and ELIC-α1GABA$_A$R.

For protein expression, the plasmid was transformed into Rosetta(DE3)pLysS (Novagen) cells under double selection with kanamycin (50 μg/mL) and chloramphenicol (35 μg/mL). Three to five isolated colonies were used to inoculate 100 mL LB media with the antibiotics and grown overnight at 37 °C in an environmental shaker at 250 rpm. The overnight culture was then diluted 1:100 into 6 × 1 L of LB media with the antibiotics and grown to an optical density of ~0.7–0.8. All six liters were harvested (5000 rpm, 20 min, 4 °C) and suspended into 2 L of LB media supplemented with 0.5 M sorbitol. The concentrated cells were equilibrated in an environmental shaker at 15 °C and 250 rpm for 1 h before inducing expression with 0.2 mM isopropyl β-D-1-thiogalactopyranoside. The cells were harvested after ~20 h expression, re-suspended in 50 mM Tris pH 8, 300 mM NaCl and flash frozen in liquid N$_2$. For protein purification, the frozen cells were thawed and homogenized with the addition of 2 mg/mL lysozyme and 1 μL benzonase before lysis using an M-110Y microfluidizer processor (Microfluidics). Membranes were pelleted by ultracentrifugation at 45 krpm at 4 °C in a Type 45Ti rotor. The fusion protein was extracted with 2% (w/v) n-dodecyl-β-D-maltoside (DDM, Anatrace), purified using a 5-mL HisTrap HP column (GE Healthcare), eluted with 250 mM imidazole, and then desalted in a buffer of 50 mM sodium phosphate at pH 8, 250 mM NaCl, and 0.05% (w/v) DDM. MBP was cleaved by TEV protease overnight and separated from ELIC-α1GABA$_A$R using a 1-mL HisTrap HP column. The pentameric fraction of ELIC-α1GABA$_A$R was collected in a buffer containing 10 mM sodium phosphate at pH 8, 150 mM NaCl, 0.05% (w/v) DDM by size exclusion chromatography using a Superdex 200 10/300GL column (GE Healthcare). The purified pentameric ELIC-α1GABA$_A$R was concentrated to ~4 mg/mL for crystallization.

**Crystallography and data analysis.** Crystals were obtained using the sitting-drop vapor diffusion method at 4 °C. All chemicals used for crystallization were purchased from Sigma-Aldrich (St. Louis, MO) unless stated otherwise. The reservoir solution contained 19–21.5% PEG 400, 400 mM NaSCN, 100 mM MES buffer at pH 6.1 and was mixed with the protein in a 1:1 ratio for crystallization of apo ELIC-α1GABA$_A$R. For co-crystallization, alphaxalone in the concentration of 10–1000 μM was mixed with the protein for ~30 min before setting up trays. The crystals were obtained typically after 3–4 weeks and cryo-protected by using up to 35% PEG400 before being flash-frozen in liquid nitrogen for storage.

The x-ray diffraction data of apo crystals were collected on the Southeast Regional Collaborative Access Team (SERCAT) beamline 22-ID at the Advanced Photon Source, Argonne National Laboratory. The data of alphaxalone-bound co-crystals were collected on the beamline 12–2 at the Stanford Synchrotron Radiation Lightsource (SSRL). The collected datasets were indexed, integrated, and scaled using the XDS program[61]. The scaled datasets were merged with Aimless[62]. Details of crystal parameters and data collection statistics for both crystals are provided in Table 1.

The molecular replacement method was used for the initial structure determination. The ECD of ELIC (PDB code: 4Z90)[63] and the TMD of β3GABA$_A$R (PDB code: 4COF)[13] were used as the isolated search ensembles for the molecular replacement solution by PHASER[64]. The TMD of the initial solution was then mutated to α1GABA$_A$R and used as the new search model for molecular replacement. The results clearly indicated two pentamers in the crystallographic asymmetric unit. This initial model was then refined in both the BUSTER[65] and Phenix[66] programs. Non-crystallographic symmetry (NCS) restraints were used throughout refinement. After the initial refinements, 2$F_O$-$F_C$ composite and simulated annealing omit maps were calculated and examined to improve the model. The entire structure was iteratively analyzed, rebuilt with the program Coot[67] and refined in Phenix[66].

The refined pentameric structure of apo ELIC-α1GABA$_A$R was used as the molecular replacement model for the structure of alphaxalone-bound crystals. The solution also contained two pentamers in the crystallographic asymmetric unit. After initial rounds of refinement, the electron density difference map ($F_O$-$F_C$) showed strong density for alphaxalone in the TMD. The initial structure of alphaxalone was obtained from a previous publication[68] and the alphaxalone molecule was fit to the $F_O$-$F_C$ difference density in Coot. The Translation-Libration-Screw-rotation model (TLS) and torsional non-crystallographic symmetry (NCS) restraints were applied to all subunits of two pentamers in the asymmetric unit and the final model was refined in Phenix[66].

The geometry and stereochemistry of the final structures were validated by the program MolProbity[69]. The refinement statistics are given in Table 1. All molecular graphics were prepared using PyMol[70] or VMD[71].

**Electrophysiology.** Functional properties of the α1GABA$_A$R chimera, its mutants, and ELIC were measured using two-electrode voltage clamp (TEVC) electrophysiology of Xenopus laevis oocytes expressing the channels of interest. All procedures involving Xenopus laevis oocytes were approved by the University of Pittsburgh Institutional Animal Care and Use Committee. A T7 promoter followed by DNA encoding a selected channel was inserted into the pCMV-mGFP Cterm S11 Neo Kan vector (Theranostech, NM). Capped complementary RNA was synthesized with the mMessage mMachine T7 kit (Ambion), purified with the RNeasy kit (Qiagen), and injected (4–25 ng) into Xenopus laevis oocytes (stages 5–6). Site-directed mutagenesis was performed using the QuickChange Lightning Kit (Agilent) and confirmed by sequencing at the University of Pittsburgh Health Sciences Genomics Research Core. Oocytes were maintained at 10 or 18 °C in modified Barth's solution containing 88 mM NaCl, 1 mM KCl, 2.4 mM NaHCO$_3$, 15 mM HEPES, 0.3 mM Ca(NO$_3$)$_2$, 0.41 mM CaCl$_2$, 0.82 mM MgSO$_4$, 10 μg mL$^{-1}$ sodium penicillin, 10 μg mL$^{-1}$ streptomycin sulfate, and 100 μg mL$^{-1}$ gentamycin sulfate at pH 6.7. Two-electrode voltage clamp experiments were performed at room temperature 1–4 days after injection with an OC-725C Amplifier (Warner Instruments) and Digidata 1440 A digitizer (Axon Instruments) in a 20-μL oocyte recording chamber (Automate Scientific). Oocytes were clamped to a holding potential of −60 mV. Oocytes expressing the α1GABA$_A$R chimera and its mutants were recorded using ND96 solution (96 mM NaCl, 2 mM KCl, 1.8 mM CaCl$_2$, 1 mM MgCl$_2$ and 5 mM HEPES at pH 7.4). The recording solution for oocytes expressing ELIC contained 130 mM NaCl, 0.1 mM CaCl$_2$, and 10 mM HEPES at pH 7.4 to minimize inhibition by divalent cations[72]. Alphaxalone stock solution was prepared in DMSO and the final DMSO concentration used for experiments was no more than 0.01%. Data were collected and processed using Clampex 10 (Molecular Devices). Non-linear regressions were performed using Prism 7.0 (Graphpad).

**Molecular dynamics simulations.** The crystal structure of the alphaxalone-bound ELIC-α1GABA$_A$R (PDB code: 6CDU) was used as initial coordinates in MD simulations. The alphaxalone-bound ELIC-α1GABA$_A$R was embedded into a pre-equilibrated lipid bilayer composed of POPC/cholesterol in a 5:1 molar ratio[73,74] using the GROMACS g_membed tool[75]. The system was solvated in TIP3P water, ionized with 100 mM NaCl, and contained ~172,000 atoms.

Three replicate 50-ns MD simulations were run using GROMACS 2016[76] and the CHARMM36 force field[77]. Alphaxalone geometry and parameters were assigned by analogy using the CHARMM General Force Field (CGenFF) for drug-like molecules[77] and refined using the Force Field Toolkit (ffTK) protocol[78]. Optimized alphaxalone structure and parameters are provided in the Supporting Materials (Supplementary Fig. 13, Supplementary Tables 3–5). The system was energy minimized for 10,000 steps with harmonic position restraints of 10 K kJ/mol/nm$^2$ on the protein backbone atoms, followed by 3 ns of equilibration, during which position restraints on the protein backbone were gradually reduced from 10,000 to 0 kJ/mol/nm$^2$. Production simulations were performed at a constant pressure and temperature (NPT) of 1 atm and 310 K with a 2-fs time step. Bonds were constrained using the LINCS algorithm[79]. The particle mesh Ewald method was used for long-range electrostatic interactions[80]. A 12-Å cutoff was used for nonbonded interactions. Full electrostatic and non-bonded interactions were evaluated every 2 fs and the neighbor list was updated every 10 fs. Systems were simulated with periodic boundary conditions in three dimensions.

## Data availability

Data supporting the findings of this manuscript are available from the corresponding author upon reasonable request. Crystal structures of apo and alphaxalone-bound α1GABA$_A$R chimeras are deposited in the Protein Data Bank with accession codes 6D1S and 6CDU, respectively.

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

## Acknowledgements

The authors thank Prof. William Furey for his help and discussion in data collection and analysis, Mr. Devin Adell for his participation in the early electrophysiology data collection, and Ms. Sandy Hirsh for her suggestions on the manuscript. Use of the Stanford Synchrotron Radiation Lightsource, SLAC National Accelerator Laboratory, is supported by the U.S. Department of Energy, Office of Science, and Office of Basic Energy Sciences under Contract No. DE-AC02-76SF00515. The SSRL Structural Molecular Biology Program is supported by the DOE Office of Biological and Environmental Research, and by the National Institutes of Health, National Institute of General Medical Sciences (including P41GM103393). Use of the Advanced Photon Source was supported by the U. S. Department of Energy, Office of Science, Office of Basic Energy Sciences, under Contract No. W-31-109-Eng-38. The MD simulations were supported in part by the National Science Foundation through XSEDE resources (TG-MCB050030N). The research was supported by NIH funding (R01GM056257). The content is solely the responsibility of the authors and does not necessarily represent the official views of the National Institutes of Health.

## Author contributions

P.T. and X.Y. designed the project. Q.C. conducted protein preparation and crystallization. Q.C., P.A., A.E.C., and P.T. contributed to x-ray data collection. Q.C. and P.A. processed diffraction data and refined the structures. Q.C., M.M.W., P.A., and P.T. analyzed structures. Q.C., T.S.T., and M.M.W. performed TEVC functional measurements and data analysis. M.M.W. and Q.C. performed molecular dynamics simulations and they, along with P.T., analyzed data. P.T., Q.C., and M.M.W. wrote the manuscript with input from other authors. All authors contributed and reviewed the results and approved the final version of the manuscript.

## Additional information

**Competing interests:** The authors declare no competing interests.

