## [Peer Review File · Nature Communications]

Reviewers' comments:

Reviewer #1 (Remarks to the Author):

Review of Chen et al.

Here, Chen et al. structurally map and functionally validate an alphaxalone binding site at an engineered GABA-A receptor $\alpha 1$ TMD – ELIC ECD chimera. They nicely demonstrate that the ELIC orthosteric agonist propylamine activates the channel and the GABA-A $\alpha 1$ TMD PAM/allosteric agonist alphaxalone potentiates and directly activates the channel. The authors present two x-ray crystallographic structures of the detergent-solubilized chimera, an Apo structure and an alphaxalone-bound structure. The authors interpret their results as having obtained physiologically-meaningful resting (Apo) and desensitized (alphaxalone-bound) states. The structural biology component is convincing in terms of the alphaxalone site, however I have concerns about the resolution cutoffs chosen (details below). The justification for the orientation chosen for modeling of the ligand in its density requires more detail, but I am firmly convinced the overall site is correct. To make this study much stronger, corresponding mutations in a physiological GABA-A receptor (alpha-beta-gamma or whatever the most relevant assembly) would be helpful- to show that the conclusions made from their elegant model system are indeed relevant to a physiological receptor assembly.

One potential weakness in this study relates to degree of confidence in conformational state assignment. The conformations of the pore-lining helices are \sim equivalent between the two structures. The authors express high confidence that their apo structure is representative of a resting state of the receptor. The authors are clearly aware, however, that if this structure is indeed representative of a physiological resting state, then this receptor is unique or almost unique among all other pentameric receptors. There is overwhelming evidence from physiology and a bit from structural biology that the resting state gate is at the 9' position at the midpoint of the pore. The authors support their state assignment logically based on physiology (channel is in a resting state when agonists are not present), differences in polarity in pore-lining residues compared to other channels, and comparison with the Torpedo nicotinic receptor. The caveats I suggest they consider include: (1) detergent and crystallization can make it difficult/impossible to resolve different conformational states (e.g. all the trouble with getting different conformations of GLIC); (2) the Torpedo receptor has a well-described register error through the M2 helix that makes the comparison in SF5 meaningless (related to that specific receptor comparison, all the others are great). I would be very happy if the authors simply expressed in the text the possibility of detergent- and/or crystallization-related artifacts on conformation. Or, there is needed some kind of functional study to support their conclusion that the resting state gate in this channel is at $\sim 2'$ or $-2'$ position (e.g., scam).

Specific comments:

1. Intro, 3rd paragraph, could a review be cited in place of refs 6-15?
2. Many figures could be improved in clarity and impact by adding some labels. For example Fig. 2, consider adding indications of the TMD and ECD and conformational state (apo, ligand-bound) to the figure panels.
3. The title and much of the overall theme of the study focus on a 'bottom up' up mechanism, which I take to mean that the ligand that binds at the base of the receptor affects what is happening in the agonist site. However, the allosteric pathway (structural linkage) and the area around the agonist site do not appear to be much if at all different in the two conformations- looking at SF7 as an example, where backbones superimpose \sim perfectly.
4. p.4, terminology of R199-K222 "covalent link" is a bit confusing. I take it to mean that R199 is the final residue from ELIC and is linked by a peptide bond to K222 as the first residue from the GABA-A $\alpha 1$ subunit. Is the final pdb model not made to contain a continuously numbered set of

amino acids? It might be simpler to have the numbering be continuous so that bonds appear properly drawn when pdb files are examined- and might be easier to understand in the text also if the numbering is continuous. Just a suggestion.

5. Please provide an Fo-Fc omit map showing density for the ligand. Could be a supplementary figure. In addition, or as an alternative, an Fo-Fo map comparing the two datasets could provide a very strong argument for the ligand and water molecule modeling.

6. What is the B-factor for the water molecule? What is the basis for not modeling it as an ion? In the higher resolution structures from the other groups with neurosteroids bound at the same/similar site, they saw no water- what is different here? The bar is high because the comparative resolution is fairly low, too low to be confident about assigning waters without corroborating evidence.

7. Related to this map figure request, what is the basis for modeling alphaxalone in the way the authors chose? Does the density and energy-minimized chemical structure allow for unambiguous modeling in a single orientation and conformation? Or could it fit a couple of different ways?

8. Final full paragraph on p. 4: several interactions are listed but it is unclear why these and not other intra and inter subunit interactions are described? What is the conclusion/relevance of these interactions specifically? Maybe this detail would fit better in another section?

9. Fig. 4, please mention distances for atomic interactions either in the text, figure or legend.

10. p. 6 and Fig. 6 legend, "indole amide", do you mean indole amine?

11. Do the mutants tested in Fig. 5c affect activation by orthosteric propylamine (does the EC50 for propylamine change)?

12. Do these mutants affect direct activation by alphaxalone, or only the potentiation of propylamine activation?

13. Table I: x-ray data processing and model refinement statistics. There are a number of issues here suggestive of resolution cutoffs being too high (absolute values for resolution cutoffs are too low); please re-process and refine or provide an explanation with map comparisons justifying inclusion of the high resolution data:

a. CC1/2 should be >0.5 , or at an absolute minimum >0.4 , or the authors need to explain why they included the higher resolution data.

b. Rpim should not be above 0.5, or not much above.

c. $I/\sigma I$ overall is very low for the ligand-bound complex (2.6 compared to >10 for apo).

d. $I/\sigma I$ in the highest resolution shell must be >1 .

e. B factors are quite high for structures at claimed resolutions of 3.4 and 3.1A.

f. Rfree for ligand-bound model is much too high, should be max 30% for a 3.4A structure.

Including a large amount of high resolution noise will increase Rfree dramatically. Likely all these problems will go away if a proper resolution cutoff is used (maybe 3.7 or 3.9A for the ligand bound structure, 3.4A for apo, just guesses).

g. Bond length and angle rmsds for ligand-bound structure are too high, suggestive of noisy data from including too much high resolution noise and too loose restraints.

h. If d*_{trek} did not report the values, it suggests (if I understand correctly) that the resolution cutoff is too high.

14. Please add Ramachandran validation to Table I, preferably from Molprobity server, including percentile rank from Molprobity.

15. Please provide at least a supplementary figure comparing the alphaxalone site with neurosteroids sites recently mapped structurally by the Smart and Aricescu groups, and discuss similarities and differences in the text. What do we learn that is new?

16. Please add a figure (supplementary is fine) showing electron density for the M1-M2 loop highlighted in Fig. 7 to give us confidence that the differences in conformation are robust.

17. The authors mention that alphaxalone is an antagonist at full-length ELIC. Do they learn anything from comparing their chimera structure with the published ELIC structures about how this diverging activity might be related to structural differences?

Reviewer #2 (Remarks to the Author):

Mechanisms underlying neurosteroids action on GABAAR are still poorly understood. In this respect, this manuscript provides new insights into binding and action of alphaxalone in a chimeric channel that includes the extracellular domain of ELIC and the transmembrane domain of $\alpha 1$ GABAAR. The chimeric channel exhibits activation by propylamine and potentiation by alphaxalone. Two crystal structures are presented one with and the other without alphaxalone. The binding site for the neurosteroids is in a pocket recently shown in $\alpha 5$ -GABAAR to also bind another neurosteroids, pregnanolone. This is an interesting study addressing a fundamentally important question on pLGIC modulation. I have some concerns over the interpretation which the authors need to address in their revision.

- The M2 pore as presented, looks identical (3b) and not convincing enough to represent two different conformational state. The difference is a tiny rotameric change in 2' position. A change that thermal motion can achieve. Why then does this channel not have constitutive activity?
- I am intrigued by the assignment of the closed state conformation. The currents in electrophysiology are very small of the order of 0.01-0.05 μ A for oocyte expression. It is possible that the major population of the channel is in a desensitized state?
- There are no structural differences between the two states at the level of the ECD? One would expect that the ECD in the alphaxalone bound state to correspond to that of high-GABA affinity state.
- The R factors for alphaxalone bound structure is a bit worrisome. Why was only 2.44% of the data used for Rfree set?
- The abstract should explicit say what type of chimera it is: ELIC- $\alpha 1$ GABAR chimera. As with all other chimeras described in the text both channel types should be included in the name (as in the supplementary figure 1). Also, please include the description of how the chimera was engineered in the results section.
- The differences between the two structures are subtle and local at the intracellular end. It would be good to show 2Fo-Fc maps of this region for both structures.
- Please show alphaxalone density as an Fo-FC map contoured at 3sigma

Reviewer #3 (Remarks to the Author):

The manuscript of Chen et al reports on two crystal structures of a new $\alpha 1$ GABA receptor chimera solved both in the apo state and in complex with the neurosteroid anesthetic alphaxalone. Since the chimera displays functional and pharmacological properties similar to pristine GABAARs, including potentiation, activation and desensitization by alphaxalone, the authors conclude that these structures provide a structural basis of neurosteroid anesthetic binding to GABAAR, which is critically important for the discovery of new GABAAR modulators with therapeutic potential. The discovery of a new GABAAR chimera that is functional without sequence optimization, activated by agonists in a concentration-dependent manner including the neurosteroids, and inhibited by picrotoxin is an interesting result. Also, the structural characterization of neurosteroid binding at high-resolution along with the identification of the key stabilizing interactions, which is well supported by electrophysiology on mutants and corresponding MD simulations, provide important insights. Overall, this study is timely and potentially influential in the field of ligand-gated ion channels. However, as the manuscript stands I have substantial concern, which I list below in order of importance. Unless, these points are clearly addressed and the manuscript revised accordingly, I cannot recommend it for publication in Nature Communications.

Major points:

First, the $\alpha 1$ GABA receptor chimera was solved in the absence of ligands and in complex with alphaxalone. Throughout the paper, these structures are referred to as representative of the resting and desensitized states of the ion channel. How can the authors come up with such a non-ambiguous functional annotation? Both arguments on the crystallization time scale relative to the

desensitization rate (page 4) and the size of the ion pore in the apo form that is too small to allow for the passage of hydrated chloride (page 5) are not very convincing. In addition, the observation that the pore profile of the putative resting state is considerably different from those reported for other pLGICs sounds as an alarm bell, which requires stronger evidence to support the claim. In this respect, the argument that GABARs have a different pattern of hydrophobicity in the pore as compared to other pLGICs (page 7) is not a valid one, as one cannot compare anionic with cationic channels. Indeed the comparison with the anionic channel GluCl in Figure S8 shows perfect conservation of hydrophobicity (by the way GlyR $\alpha 1$ and GlyR $\alpha 3$ should be added to the multiple sequence alignment in Figure S8). In my view, the structure of the putative resting state (i.e. apo $\alpha 1$ GABAR chimera) could be representative of one desensitized state. Perhaps, an analysis of the binding affinity of alphaxalone to the apo versus the alphaxalone-bound forms in simulation might provide insights.

Second, the authors claim that the structural comparison of apo versus alphaxalone-bound forms provide new mechanistic insights into the process of activation and desensitization by neurosteroid binding. Provided that the functional annotation of structures is correct, which is highly debatable (see above), this comparison may provide, at best, an illustration of the conformational changes associated with neurosteroid binding with no sequential order nor link of causality between them. Therefore, there is no evidence supporting the "bottom-up" mechanism for channel activation and desensitization by alphaxalone, which is referred to as "trajectory of conformational changes" on page 8. This is an over claim and should be retracted.

Third, the structural characterization of neurosteroid binding in GABAR is not that novel. The recent structural analysis of Miller et al Nat Struct Mol Biol 2017 (Ref.15) considerably reduces the potential impact of the alphaxalone-bound structure presented here.

Fourth, beside the configuration of the ion pore, which is carefully analyzed, little structural information is provided on the comparison between the apo and alphaxalone-bound forms. Specifically no information is provided on the quaternary structure of the chimeric receptor, which is supposed to undergo substantial conformational change (i.e. global twisting and blooming) on activation and desensitization, nor on the conformations of the orthosteric neurotransmitter site and the inter-subunit allosteric sites in these structures, which are supposed to modulate the binding affinity for the agonist(s) on gating (resting to active) and desensitization. This important information along with a comparison with existing structures of GABAR should be added to the second subsection of the "Results".

Fifth, the third subsection of the "Results" is very confusing. By comparing the structure of the $\alpha 1$ -GABAR chimera in the apo state with that of the "parental" ELIC (also in the apo state), the authors conclude that "both structural cooperation and independence (of the ECD and TMD) contribute to the assembly of functional chimeric channels". Beside the fact that this sentence makes little sense (either there is cooperation or not), how can one draw meaningful conclusions by comparing structures of possibly non-equivalent states? And most importantly, is this comparison useful to the economy of the paper? I acknowledge that the crystallographic evidence that the conserved lysine on the M2-M3 loop (K279) makes an inter-subunit H-bond with T28 on the $\beta 1$ - $\beta 2$ loop, as observed in the X-ray structure of $\beta 3$ -GABAR, is an interesting observation that should be mentioned. Everything else seems to go beyond the scope of the paper.

Minor points:

1. A structural analysis of the MD simulation trajectories, e.g. the time series of the overall RMSD, TMD, ECD etc., is missing.
2. Some of the figures, particularly Figure 2 and 4, are unclear and could be improved.

Reviewer #1

The structural biology component is convincing in terms of the alphaxalone site, however I have concerns about the resolution cutoffs chosen (details below). The justification for the orientation chosen for modeling of the ligand in its density requires more detail, but I am firmly convinced the overall site is correct.

Both issues are addressed in the sections below.

To make this study much stronger, corresponding mutations in a physiological GABA-A receptor (alpha-beta-gamma or whatever the most relevant assembly) would be helpful- to show that the conclusions made from their elegant model system are indeed relevant to a physiological receptor assembly.

We have added the following paragraph to the Results section:

“These findings from the $\alpha 1$ GABA_AR chimera are consistent with the results from the full-length GABA_ARs. A number of previous studies on full-length GABA_ARs show that the $\alpha 1$ -W246L and $\alpha 1$ -Q242L mutations significantly suppressed allopregnanolone potentiation^{47,48}. Similarly, alphaxalone potentiation of $\alpha 1\beta 3$ GABA_AR is also significantly reduced in the $\alpha 1$ -T306A mutant compared to the WT receptor (Fig. S7).”

One potential weakness in this study relates to degree of confidence in conformational state assignment. The conformations of the pore-lining helices are ~equivalent between the two structures. The authors express high confidence that their apo structure is representative of a resting state of the receptor. The authors are clearly aware, however, that if this structure is indeed representative of a physiological resting state, then this receptor is unique or almost unique among all other pentameric receptors. There is overwhelming evidence from physiology and a bit from structural biology that the resting state gate is at the 9' position at the midpoint of the pore. The authors support their state assignment logically based on physiology (channel is in a resting state when agonists are not present), differences in polarity in pore-lining residues compared to other channels, and comparison with the Torpedo nicotinic receptor. The caveats I suggest they consider include: (1) detergent and crystallization can make it difficult/impossible to resolve different conformational states (e.g. all the trouble with getting different conformations of GLIC); (2) the Torpedo receptor has a well-described register error through the M2 helix that makes the comparison in SF5 meaningless (related to that specific receptor comparison, all the others are great). I would be very happy if the authors simply expressed in the text the possibility of detergent- and/or crystallization-related artifacts on conformation. Or, there is needed some kind of functional study to support their conclusion that the resting state gate in this channel is at ~2' or -2' position (e.g., scam).

We have taken the reviewer's suggestion and removed the discussion about the *Torpedo* receptor. It was also to our initial surprise that the “gate” was not at the 9' position. After researching the database, we noticed that GluCl is the only anion-conducting channel whose resting-state structure has been solved. It has a pore hydrophobicity pattern similar to a number of cation pLGICs. In addition to the discussion about the pore hydrophobicity patterns, we have added the following to the Discussion:

“Because of the unconventional occlusion location in the pore of the apo ELIC- $\alpha 1$ GABA_AR, it is natural to question whether the apo structure reflects a true resting-state conformation of ELIC- $\alpha 1$ GABA_AR. We have analyzed the question from several different angles. Could the DDM

detergent and crystallization conditions have introduced conformational biases to the resting ELIC- α 1GABA_AR? DDM has been used for purification and crystallization of many different pLGICs, including GluCl1 that exhibits different pore conformations for the resting and desensitized channels³⁴. Both ELIC and ELIC- α 1GABA_AR are crystallized with DDM, but show distinctly different pore conformations in the resting-state condition. Thus, it is not convincing to simply attribute the unconventional resting-state pore profile to detergent- and/or crystallization-related artifacts, even though one cannot completely rule out this possibility. Can we assign the apo ELIC- α 1GABA_AR to a “desensitized” state in the absence of agonist because of the similar structures of the apo and desensitized ELIC- α 1GABA_AR? The classic definition of desensitization requires the presence of agonists or channel modulators. Moreover, our electrophysiology functional data also do not support this assignment. Can a resting-state GABA_AR adopt the non-conducting channel conformation similar to those observed for desensitized pLGICs? Our apo structure suggests such a possibility. In addition, this possibility has been suggested previously by a SCAM study⁴⁴, in which the charged sulfhydryl reagents were able to penetrate from the ECD of α 1 β 1 γ 2GABA_AR to the pore at the level of α 1V257 (2') in the resting state. We also notice that GABA_ARs have a different pattern of hydrophobicity in pore-lining residues compared to those pLGICs with known resting structures (**Fig. S11**). The anion-conducting GluCl³⁴ along with the cation-conducting 5HT₃AR^{36,38}, *Torpedo* nAChR⁶², GLIC⁴³, and ELIC^{23,42} have consecutive hydrophobic rings at the 9', 13' and 17' (16') positions. In contrast, GABA_ARs have mostly hydrophilic pore-lining residues except at the 9', -2' and 2' positions (Fig. S11). Combining all aforementioned information with the pore radius profiles for the resting and desensitized α 1GABA_AR chimera, we propose a channel gate for both activation and desensitization at the **cytoplasmic** end of the pore formed by the 2' and -2' residues. Whether the finding can be generalized to other GABA_AR mandates additional structural investigations, particularly with heteromeric GABA_AR.”

In the Results section, we also have added the following:

“a previous study of α 1 β 1 γ 2GABA_AR using the scanning-cysteine-accessibility method (SCAM) reported that, in the absence of GABA, charged sulfhydryl reagents applied from the extracellular end of the resting-state channel were able to penetrate to the level of α 1V257 (2')⁴⁴, suggesting a similar pore profile to that shown in the structure of the apo α 1GABA_AR chimera (Fig. 3).”

Specific comments:

1. *Intro, 3rd paragraph, could a review be cited in place of refs 6-15?*

All citations are representative works in the field, from *in vivo* to the molecular level and from general anesthetics to neurosteroids. No single review article has covered all of them.

2. *Many figures could be improved in clarity and impact by adding some labels. For example Fig. 2, consider adding indications of the TMD and ECD and conformational state (apo, ligand-bound) to the figure panels.*

Most figures have been revised to improve clarity.

3. *The title and much of the overall theme of the study focus on a ‘bottom up’ up mechanism, which I take to mean that the ligand that binds at the base of the receptor affects what is happening in*

the agonist site. However, the allosteric pathway (structural linkage) and the area around the agonist site do not appear to be much if at all different in the two conformations- looking at SF7 as an example, where backbones superimpose ~perfectly.

Alphaxalone binding at the base of the receptor introduces 0.84 Å RMSD to the orthosteric agonist binding site at loop C (D172-N186). We include the data in the revised manuscript.

4. *p.4, terminology of R199-K222 “covalent link” is a bit confusing. I take it to mean that R199 is the final residue from ELIC and is linked by a peptide bond to K222 as the first residue from the GABA-A $\alpha 1$ subunit. Is the final pdb model not made to contain a continuously numbered set of amino acids? It might be simpler to have the numbering be continuous so that bonds appear properly drawn when pdb files are examined- and might be easier to understand in the text also if the numbering is continuous. Just a suggestion.*

To prevent confusion, we edited the sentence to the following:

“In addition to the peptide bond linking the last ECD residue R199 of ELIC to the first TMD residue K222 of $\alpha 1$ GABA_AR ...”

In order to provide convenience to the readers and make it easier to compare the results from the chimeras with original GABA_ARs, we need to maintain the standard residue numbering for the $\alpha 1$ GABA_AR TMD. Thus, there is a gap in the numbering of ECD and TMD residues.

5. *Please provide an Fo-Fc omit map showing density for the ligand. Could be a supplementary figure. In addition, or as an alternative, an Fo-Fo map comparing the two datasets could provide a very strong argument for the ligand and water molecule modeling.*

See new Fig. 2. Since the 2Fo-Fc map for alphaxalone is shown in Figure 4b, we replaced the 2Fo-Fc maps with the Fo-Fc omit map at 4σ in the new Fig. 2.

6. *What is the B-factor for the water molecule? What is the basis for not modeling it as an ion? In the higher resolution structures from the other groups with neurosteroids bound at the same/similar site, they saw no water- what is different here? The bar is high because the comparative resolution is fairly low, too low to be confident about assigning waters without corroborating evidence.*

We did not fit the observed Fo-Fc density (old Fig. 6a) to a water or ion molecule because of the limited resolution. Independently, in our MD simulations, we saw a water molecule close to alphaxalone in the position that almost perfectly coincides with the Fo-Fc omit electron density map observed in the alphaxalone-bound structure. However, considering the limited resolution and that not every equivalent site shows this Fo-Fc density, we have decided to remove the result and discussion of water in the revised manuscript.

7. *Related to this map figure request, what is the basis for modeling alphaxalone in the way the authors chose? Does the density and energy-minimized chemical structure allow for unambiguous modeling in a single orientation and conformation? Or could it fit a couple of different ways?*

As shown in new Fig. 2, the Fo-Fc omit maps are unambiguous for alphaxalone. The shape of the omit map suggests three possibilities for modeling alphaxalone, 1) the current way, 2) rotate the current molecule along long axis; or 3) flip the molecule 180 degrees over the long axis. We eliminated possibility 2) because it causes alphaxalone to clash with W246 and destroys the stacking interaction. We eliminated possibility 3) for two reasons: this removes hydrogen bonding

and causes the alphaxalone C20 methyl group to clash with Q242. Thus, alphaxalone is modelled in the current orientation that favors both stacking and hydrogen bonding interactions without clashing.

8. *Final full paragraph on p. 4: several interactions are listed but it is unclear why these and not other intra and inter subunit interactions are described? What is the conclusion/relevance of these interactions specifically? Maybe this detail would fit better in another section?*

We have edited the paragraph to clarify this point. The ECD-TMD interactions vital for functional channels suggested by previous studies are also observed in ELIC- $\alpha 1$ GABA_AR.

9. *Fig. 4, please mention distances for atomic interactions either in the text, figure or legend.*

We have added the distances either to the figures or in the legends.

10. *p. 6 and Fig. 6 legend, “indole amide”, do you mean indole amine?*

This has been corrected.

11. *Do the mutants tested in Fig. 5c affect activation by orthosteric propylamine (does the EC₅₀ for propylamine change)?*

12. *Do these mutants affect direct activation by alphaxalone, or only the potentiation of propylamine activation?*

These questions are answered by the data shown in the new Fig. 4. The related text reads as following:

“Two of these mutants, Q242L and T306A, are similar to the WT $\alpha 1$ GABA_AR chimera in their response to the orthosteric agonist propylamine, whereas the EC₅₀ of the W246L mutant is increased two orders of magnitude compared to WT (**Fig. 4c**), signaling the importance of this conserved residue W246 in the function of GABA_ARs. For the Q242L and T306A mutants, 0.1 μ M alphaxalone still potentiates channel currents but with smaller magnitudes compared to the WT chimera (**Fig. 4d**). The W246L mutation has a stronger effect and almost completely abolishes potentiation by the same concentration of alphaxalone (**Fig. 4d**). As expected, these mutations also reduce channel activation by alphaxalone (**Fig. 4e**). These findings from the $\alpha 1$ GABA_AR chimera are consistent with the results from the full-length GABA_ARs. A number of previous studies on full-length GABA_ARs show that the $\alpha 1$ -W246L and $\alpha 1$ -Q242L mutations significantly suppressed allopregnanolone potentiation^{47,48}. Similarly, alphaxalone potentiation of $\alpha 1\beta 3$ GABA_AR is also significantly reduced in the $\alpha 1$ -T306A mutant compared to the WT receptor (**Fig. S7**).”

13. *Table I: x-ray data processing and model refinement statistics. There are a number of issues here suggestive of resolution cutoffs being too high (absolute values for resolution cutoffs are too low); please re-process and refine or provide an explanation with map comparisons justifying inclusion of the high-resolution data...*

a. *CC1/2 should be >0.5, or at an absolute minimum >0.4, or the authors need to explain why they included the higher resolution data.*

b. *Rpim should not be above 0.5, or not much above.*

c. *I/sigmaI overall is very low for the ligand-bound complex (2.6 compared to >10 for apo).*

d. *I/sigmaI in the highest resolution shell must be >1.*

e. *B factors are quite high for structures at claimed resolutions of 3.4 and 3.1A.*

f. Rfree for ligand-bound model is much too high, should be max 30% for a 3.4Å structure. Including a large amount of high resolution noise will increase Rfree dramatically. Likely all these problems will go away if a proper resolution cutoff is used (maybe 3.7 or 3.9Å for the ligand bound structure, 3.4Å for apo, just guesses).

g. Bond length and angle rmsds for ligand-bound structure are too high, suggestive of noisy data from including too much high resolution noise and too loose restraints.

h. If d*trek did not report the values, it suggests (if I understand correctly) that the resolution cutoff is too high.

The newly processed apo structure and newly collected alphaxalone-bound structure have addressed these issues.

14. Please add Ramachandran validation to Table I, preferably from Molprobity server, including percentile rank from Molprobity.

We now include both the Ramachandran parameters and percentile rank from Molprobity in Table 1.

15. Please provide at least a supplementary figure comparing the alphaxalone site with neurosteroids sites recently mapped structurally by the Smart and Aricescu groups, and discuss similarities and differences in the text. What do we learn that is new?

See Figure S12.

16. Please add a figure (supplementary is fine) showing electron density for the M1-M2 loop highlighted in Fig. 7 to give us confidence that the differences in conformation are robust.

The new Fig 6 shows the electron density for the TM1-TM2 loop.

17. The authors mention that alphaxalone is an antagonist at full-length ELIC. Do they learn anything from comparing their chimera structure with the published ELIC structures about how this diverging activity might be related to structural differences?

Because we did not observe alphaxalone binding to the ECD in the crystal structure of the ELIC-GABAAR chimera, we suspect that alphaxalone, most likely, binds to the TMD of ELIC. However, it requires another full investigation that is beyond the scope of the current study to determine the binding location and why the binding introduces inhibition to ELIC.

Reviewer #2

1. The M2 pore as presented, looks identical (3b) and not convincing enough to represent two different conformational state. The difference is a tiny rotameric change in 2' position. A change that thermal motion can achieve. Why then does this channel not have constitutive activity?

We have expanded the structural analysis to present structural differences other than the pore region between apo and alphaxalone-bound chimeras (Fig. S5, Fig. 6, and Fig. S9). Similar pore profiles between the apo and alphaxalone-bound channels do not imply that the apo channel is desensitized or should have constitutive activity. It has not been proven that a channel in the resting state cannot adopt a non-conducting conformation similar to that observed in the desensitized pLGICs.

2. I am intrigued by the assignment of the closed state conformation. The currents in electrophysiology are very small of the order of 0.01-0.05 μA for oocyte expression. It is possible that the major population of the channel is in a desensitized state?

We have a lengthy discussion about the issue in the revised manuscript. By the conventional definition of desensitization, a channel without ligand binding cannot be viewed as desensitized.

3. There are no structural differences between the two states at the level of the ECD? One would expect that the ECD in the alphaxalone bound state to correspond to that of high-GABA affinity state.

Alphaxalone binding at the base of the receptor introduces 0.84 Å RMSD to the orthosteric agonist binding site, showing small inward movement of loop C (D172-N186). We include the data in the revised manuscript. Also see Fig. S5 and Fig S9.

4. The R factors for alphaxalone bound structure is a bit worrisome. Why was only 2.44% of the data used for Rfree set?

This has been corrected. ~5% of the data was used for Rfree set of the new alphaxalone-bound structure.

5. The abstract should explicitly say what type of chimera it is: ELIC- α 1GABAR chimera. As with all other chimeras described in the text both channel types should be included in the name (as in the supplementary figure 1). Also, please include the description of how the chimera was engineered in the results section.

Done.

6. The differences between the two structures are subtle and local at the intracellular end. It would be good to show 2Fo-Fc maps of this region for both structures.

New Fig 6 shows the electron density for the TM1-TM2 loop.

7. Please show alphaxalone density as an Fo-FC map contoured at 3sigma. Fig. 2 shows Fo-FC maps at 4 σ for alphaxalone molecules.

Reviewer #3 (Remarks to the Author):

Major points:

1. First, the α 1GABA receptor chimera was solved in the absence of ligands and in complex with alphaxalone. Throughout the paper, these structures are referred to as representative of the resting and desensitized states of the ion channel. How can the authors come up with such a non-ambiguous functional annotation? Both arguments on the crystallization time scale relative to the desensitization rate (page 4) and the size of the ion pore in the apo form that is too small to allow for the passage of hydrated chloride (page 5) are not very convincing. In addition, the observation that the pore profile of the putative resting state is considerably different from those reported for other pLGICs sounds as an alarm bell, which requires stronger evidence to support the claim. In this respect, the argument that GABARs have a different pattern of hydrophobicity in the pore as compared to other pLGICs (page 7) is not a valid one, as one cannot compare anionic with cationic channels. Indeed, the comparison with the anionic channel GluCl in Figure S8 shows perfect conservation of hydrophobicity (by the way GlyR α 1 and GlyR α 3 should be added to the multiple sequence alignment in Figure S8). In my view, the structure of the putative resting state (i.e. apo

α1GABAR chimera) could be representative of one desensitized state. Perhaps, an analysis of the binding affinity of alphaxalone to the apo versus the alphaxalone-bound forms in simulation might provide insights.

The electrophysiology data support the assignment of a desensitized state to the alphaxalone-bound GABA_AR chimera. The location of the “gate” in desensitized channels has been well accepted in the field after a number of desensitized channel structures were published. Our alphaxalone-bound structure presents the expected “gate” for the desensitized channel. We don’t see any suitable assignment for the alphaxalone-bound structure other than desensitized.

The reviewer’s main concern probably lies in our functional annotation for the apo structure. To our knowledge, all published structures for **apo** pLGICs (GluCl, 5HT₃AR, *Torpedo* nAChR, ELIC, and GLIC at neutral pH) are annotated as resting-state structures. Among them, GluCl is the only anion-conducting channel. No apo GlyR structure has been reported (all published GlyR closed channel structures are bound with ligands). Clearly, we need more structures of apo pLGICs, especially for anion-conducting GABA_ARs and GlyRs, in order to build a comprehensive perspective on the location of the channel gate.

Is it possible that some pLGICs in the resting state adopt a non-conducting channel conformation similar to that of the desensitized state? The apo ELIC-α1GABA_AR structure seems to support such a possibility.

Can we annotate the apo ELIC-α1GABA_AR to a desensitized state solely based on the similar appearance of its pore profile to a desensitized pore profile? The answer is no, otherwise it is against the original definition of a desensitized state. Following the same principle, the apo ELIC-α1GABA_AR without stimuli is in a resting state. We add a lengthy discussion in the revised manuscript regarding to whether the structure of the apo chimera represents the resting state channel.

2. Second, the authors claim that the structural comparison of apo versus alphaxalone-bound forms provide new mechanistic insights into the process of activation and desensitization by neurosteroid binding. Provided that the functional annotation of structures is correct, which is highly debatable (see above), this comparison may provide, at best, an illustration of the conformational changes associated with neurosteroid binding with no sequential order nor link of causality between them. Therefore, there is no evidence supporting the “bottom-up” mechanism for channel activation and desensitization by alphaxalone, which is referred to as “trajectory of conformational changes” on page 8. This is an over claim and should be retracted. An illustration of the conformational changes associated with binding of agonist, antagonist, or allosteric modulators has been a common practice for extracting the underlying mechanisms of channel activation, gating and desensitization {Du, 2015 #37}{Althoff, 2014 #32;Hibbs, 2011 #31}{Sauguet, 2014 #39}.

We do not intend to present a sequential order. However, it is unambiguous that the structural change is triggered by alphaxalone binding to the bottom of the TMD. The electrophysiology results show that the alphaxalone binding activates and subsequently desensitizes the channel. It is also clear that the structural changes go beyond the base of the TMD (see the Results section under “**A plausible path for channel activation and desensitization by alphaxalone**”). All of

these data support a “bottom-up” mechanism. However, to prevent unnecessary confusion, we have removed the phrase “trajectory of conformational changes”.

3. *Third, the structural characterization of neurosteroid binding in GABAR is not that novel. The recent structural analysis of Miller et al Nat Struct Mol Biol 2017 (Ref.15) considerably reduces the potential impact of the alphaxalone-bound structure presented here.*

We respectfully disagree with the reviewer’s opinion. The manuscript presents a different neurosteroid in a different GABA_AR chimeric channel compared with previously reported works.

4. *Fourth, beside the configuration of the ion pore, which is carefully analyzed, little structural information is provided on the comparison between the apo and alphaxalone-bound forms. Specifically no information is provided on the quaternary structure of the chimeric receptor, which is supposed to undergo substantial conformational change (i.e. global twisting and blooming) on activation and desensitization, nor on the conformations of the orthosteric neurotransmitter site and the inter-subunit allosteric sites in these structures, which are supposed to modulate the binding affinity for the agonist(s) on gating (resting to active) and desensitization. This important information along with a comparison with existing structures of GABAR should be added to the second subsection of the “Results”.*

We have included new figures (Fig. S5, Fig. 6, and Fig. S9) and new text to address these concerns.

“Global twisting and blooming movements of pLGICs have been proposed to accompany functional conformation changes^{34,43}. No blooming is observed between the apo and alphaxalone-bound ELIC- α 1GABA_AR. Relative to the aligned structure of the apo ELIC- α 1GABA_AR, the alphaxalone-bound structure shows a small counterclockwise twist of the ECD and a clockwise twist of the TMD around the pore axis (**Fig. S5**). This direction of twist leads toward channel opening based on a structure survey of pLGICs under different functional states^{34,39,43}. In the case of ELIC- α 1GABA_AR, however, the magnitude of the twist towards an open channel is small in the desensitized alphaxalone-bound structure with respect to the apo structure, consistent with the fact that both apo and alphaxalone-bound channels are closed (Fig. 3).”

“What is the structural basis for channel activation and desensitization by alphaxalone? In addition to the small twist movement of the ECD and TMD along the channel axis (Fig. S5), we notice a set of “bottom-up” conformational changes in the aligned x-ray structures of the apo and alphaxalone-bound ELIC- α 1GABA_AR (**Fig. 6, Fig. S9**). Alphaxalone binding to the bottom of the TMD introduces structural changes, including the orientation of W246 in the TM1, presumably to optimize its interaction with alphaxalone (Fig. 6). The change passes to the TM1-TM2 linker and further to the TM2, particularly at P253 (-2') and V257 (2'), resulting in a backbone RMSD of 0.95 Å for the residues covering from W246 in TM1 to V257 (2') in TM2 (Fig. 6b). The structural changes further propagate up to the ECD-TMD interface and other ECD regions, as evidenced by the observed RMSDs: 0.66 Å for the TM2-TM3 loop (P278 - T284), 0.61 Å for the pre-TM1 (R199-I223), 0.58 Å for the β 1- β 2 linker (V26-E30), 0.63 Å for the “Cys”-loop (N112-F126), 1.01 Å for loop A (N80-S84), and 0.84 Å for loop C (D172-N186) (**Fig. S9**). Although these small changes do not necessarily reflect the actual magnitude of the structural changes involved in the functional transitions from the resting to activated and subsequent desensitized states, they suggest a plausible “bottom-up” path for alphaxalone-induced channel activation and desensitization through the allosteric binding site in the TMD.”

5. Fifth, the third subsection of the “Results” is very confusing. By comparing the structure of the $\alpha 1$ -GABAR chimera in the apo state with that of the “parental” ELIC (also in the apo state), the authors conclude that “both structural cooperation and independence (of the ECD and TMD) contribute to the assembly of functional chimeric channels”. Beside the fact that this sentence makes little sense (either there is cooperation or not), how can one draw meaningful conclusions by comparing structures of possibly non-equivalent states? And most importantly, is this comparison useful to the economy of the paper? I acknowledge that the crystallographic evidence that the conserved lysine on the M2-M3 loop (K279) makes an inter-subunit H-bond with T28 on the $\beta 1$ - $\beta 2$ loop, as observed in the X-ray structure of $\beta 3$ -GABAR, is an interesting observation that should be mentioned. Everything else seems to go beyond the scope of the paper. As noted by this reviewer, we compared the ECD structures of apo ELIC- $\alpha 1$ GABA_AR vs. apo ELIC, which hold the same functional state and identical ECD sequences. We do not see an issue with comparing this pair of structures.

We have revised this subsection to improve the clarity and moved it to the end of the Results section. For years, our own and other’s research have suggested that pLGICs are modular proteins. The ECD and TMD work cooperatively to form functional channels; at the same time, these domains can work independently to maintain their pharmacological and functional properties in a chimeric form. Structural independence of individual domains is the foundation for making functional chimeric channels that retain the pharmacology and functions of pristine channels. The results presented in the manuscript demonstrate that the $\alpha 1$ GABA_AR TMD can retain the function of the parent channel independent of the presence of the original ECD.

The aforementioned independence of individual domains does not conflict with structural cooperativity. To us, it is two sides of the same coin. If there is no structural influence from the TMD, one would see the same ECD structure in both apo ELIC- $\alpha 1$ GABA_AR and apo ELIC. Our structural comparison, however, provides evidence showing the coupling at the ECD-TMD interface and the structural cooperation when the TMD is changed (Fig. 7, Fig. S10).

Since both structural independence and cooperation are well demonstrated in our results, we feel obligated to bring these results to our readers and help their understanding of how individual domains work cooperatively and independently in pLGICs.

The new writing reads:

“The structural independence of individual domains in pLGICs and cooperativity between these domains are essential for constructing functional chimeric channels^{14,15,49}. In this study, ELIC- $\alpha 1$ GABA_AR provides a window into how the ECD and TMD both retain their individuality in the chimera and cooperate structurally to form a functional channel. Crystal structures of apo ELIC- $\alpha 1$ GABA_AR and apo ELIC (aligned along their common ECDs) show that even under the same ECD, the chimera TMD independently adopts the $\alpha 1$ GABA_AR structure that is significantly different from the ELIC TMD structure (**Fig. 7a, b**). The ECD influence on the TMD structure is rather limited. Indeed, even an isolated TMD of pLGICs can form functional channels that retain some characteristics of the parent channels^{50,51,52}. However, the structural cooperativity between the individual domains is also observed (**Fig. 7**). Although most parts of the ECD in the two structures overlap well, there is an overall ECD

backbone RMSD of 2.1 Å, resulting mainly from the displacement of several key regions in the chimera (**Fig. 7c**). The subtle and profound displacements of the respective pre-TM1 and the TM2-TM3 loop in the chimera (Fig. 7b) may have led to the conformational adjustments of the “Cys”-loop and β 1- β 2 linker at the ECD-TMD interface, as well as alterations further up at loop A and loop C (Fig. 7c). As an example of structural cooperation between the ECD and TMD, the conserved residue K279 in the TM2-TM3 of GABA_ARs and its equivalent residue R255 in ELIC are oriented in different directions so that they form respective inter- and intra-subunit polar interactions with T28 in the β 1- β 2 linker (**Fig. S10**). The inter-subunit K279-T28 interactions would not be possible without the β 1- β 2 linker relocation in the chimera. An equivalent inter-subunit polar interaction (K274-E52) is also observed in the crystal structure of β 3GABA_AR¹⁶. The importance of this lysine residue in channel gating of GABA_ARs has been well recognized^{40,41}. Mutation of this lysine impairs function of GABA_ARs⁴¹ and is associated with epilepsy^{53,54}.”

Minor points:

1. A structural analysis of the MD simulation trajectories, e.g. the time series of the overall RMSD, TMD, ECD etc., is missing.

MD trajectories are now presented in Fig. S8.

2. Some of the figures, particularly Figure 2 and 4, are unclear and could be improved. These figures have been modified and improved.

Reviewers' comments:

Reviewer #1 (Remarks to the Author):

The authors addressed some of my concerns/requests but not all, and overall the manuscript remains too speculative, in my opinion.

1. Re-refinement improved the structural statistics considerably and these are now acceptable.
2. New citations of mutants that support structure-based conclusions are good.
3. Why do the authors not consider the strychnine-bound GlyR and apo states to be expected to be all likely representative of bona fide resting states? An antagonist-bound state should be a good model for the resting state. That is the structure that, in my opinion, is the best model we have for a resting state in the superfamily (and the newest 5HT3A structure).
4. I agree with reviewer 3 that the concept of 'bottom up' modulation is not well supported by the extremely high degree of similarity between the two models. Further I do not understand what is gained by comparisons with the full length ELIC receptor structure, comparing the ECD conformations. I am not comfortable with the bottom up idea in the title or strong conclusions about it in the manuscript.
5. I feel the authors are overall too confident about conformational state assignment of their apo conformation. I think all 3 referees share this fundamental concern. To me, calling it an apo conformation or apo state would be okay. Calling it a resting state, which is inconsistent with the broad consensus is of what a resting state pore should look like, is too strong.
6. The Fo-Fc map shown for the ligand is in a color that makes it very hard to see, and is too small a part of the figure, and does not show how the ligand is modeled into it. Please make a panel where the full panel is occupied by this map, and shows the ligand in the map, to demonstrate support confidence in modeling.

Reviewer #2 (Remarks to the Author):

The authors have addressed my concerns adequately. I do not have any further comments.

Reviewer #3 (Remarks to the Author):

In the revised manuscript, Chen et al introduced several changes and addenda in response to my criticism. Unfortunately, some of these additions do not do a good job as the manuscript is still permeated of unjustified and confusing statements that should be seriously revised or retracted. Despite I am convinced that the main result of the paper is worth publishing, I expect the authors to take my comments more seriously this time.

1. Much confusion arises from the annotation of "resting state" to a structure that presents a closed ion pore and was solved in the absence of ligands (apo), which is NOT sufficient for a non-ambiguous functional assignment as acknowledged by the authors in the Discussion. Therefore, to remove the ambiguity I request all statements below to be modified accordingly.

- Page 2 (abstract) : "Here, we report crystal structures of a new α 1GABAAR chimera (ELIC- α 1GABAAR) in the resting and desensitized states."

- Page 2 (abstract) : "The resting-state structure reveals an unconventional activation gate at the intracellular end of the pore"

- Page 3 (end of introduction) : "Here, we report crystal structures of a new α 1GABAAR chimera in the resting state."

- Page 4 (results) : "We determined x-ray structures of the α 1GABAAR chimera in the resting and desensitized states"

- Page 7 (discussion) : "We have determined both the resting and desensitized structures for ELIC-GABA chimera"

2. There is no evidence in the manuscript supporting the "bottom-up" mechanism for channel activation and desensitization by alphaxalone. The authors simply assume an induced-fit model with alphaxalone binding being the initiating event. Since there is no evidence for it and no crystal structure of any endpoint can provide information on the transition pathway or the sequence of structural events, I request the "bottom-up" keyword to be amended from the title and throughout the Text.

3. The comparison of the quaternary structure of the apo chimera versus the alphaxalone-bound state (Fig S5) is enlightening and indicates that, despite the claims on page 5, there is essentially no change in the global receptor twisting or blooming. Both twisting and blooming angles in the two chimera structures should be quantified and reported (see Martin et al. PLoS Comp Biol 2017 for an operational definition) and compared to the functional changes observed in the X-ray structures (Hibbs & Gouaux 2012 and Althoff et al. 2014) and simulations (Martin et al. PLoS Comp Biol 2017) of GluCl. Based on the results, the discussion on page 5 need to be corrected. Also, the review of Cecchini and Changeux (Neuropharmacology 2015) should be used as a reference for the functional implications of twisting/blooming on gating.

4. Given the strong residue conservation in the -2' to 13' region between GABA and GluCl, the argument that differences in the hydrophobicity pattern in the pore-lining residues produce an upward shift of the constriction point (i.e. to position 2') in the apo chimera (page 8) is not very convincing. Without further arguing, I request the author to include the anionic GlyR α 1 and GlyR α 3 in the sequence alignment in Fig S11 and leave the reader the opportunity to judge the quality of their argument.

Reviewers' comments:

Reviewer #1 (Remarks to the Author):

The authors addressed some of my concerns/requests but not all, and overall the manuscript remains too speculative, in my opinion.

We very much appreciate the comments from this reviewer and attempted to address all specific comments in the previous revision. Concerns raised in this iteration have been addressed as follows.

1. Re-refinement improved the structural statistics considerably and these are now acceptable.

Thank you.

2. New citations of mutants that support structure-based conclusions are good.

Thank you.

3. Why do the authors not consider the strychnine-bound GlyR and apo states to be expected to be all likely representative of bona fide resting states? An antagonist-bound state should be a good model for the resting state. That is the structure that, in my opinion, is the best model we have for a resting state in the superfamily (and the newest 5HT3A structure).

In our humble opinion, the resting-state pLGICs should be in an apo state by definition. In the absence of an apo GlyR structure for comparison, it is uncertain whether an antagonist-bound state (strychnine-bound GlyR) is a good model for the apo resting state.

4. I agree with reviewer 3 that the concept of 'bottom up' modulation is not well supported by the extremely high degree of similarity between the two models. Further I do not understand what is gained by comparisons with the full length ELIC receptor structure, comparing the ECD conformations. I am not comfortable with the bottom up idea in the title or strong conclusions about it in the manuscript.

The “top-down” activation process initiated by agonist binding to the extracellular domain has been well accepted, even though the details of the pathway are still a topic of discussion.

Here, our functional data (see Fig. 4, the mutation of W246L completely diminishes the channel potentiation and activation by alphaxalone), along with the structural data, show unambiguously that the channel is activated purely by alphaxalone bound to the bottom of the TMD – there is no ligand binding to other regions of the channel. This is where the phrase of “bottom-up” came from.

At the reviewers' request, we have removed the phrase “bottom-up” from the title and text.

The comparison of the same ECD in both ELIC and the chimera demonstrates structural cooperativity and independency of two different domains. It also shows how changes in the TMD affect the ECD. The comparison is relevant to the chimeric construct and the allosteric action originating at the bottom of the TMD.

5. I feel the authors are overall too confident about conformational state assignment of their apo conformation. I think all 3 referees share this fundamental concern. To me, calling it an apo conformation or apo state would be okay. Calling it a resting state, which is inconsistent with the broad consensus is of what a resting state pore should look like, is too strong.

For ligand-gated ion channels, an apo state without leaking currents is by definition in a resting state. Our electrophysiology data (Fig. 1, Figs. 4c, d, e) support the assignment of the resting state – the apo channel has no leaking current, it can be activated by agonists, and it can be activated and potentiated by alphaxalone. These are experimental facts.

There is no convincing experimental evidence (too few true apo structures) or theoretical basis to conclude that all resting-state pLGICs must uniformly adopt the same closed-channel conformation with the pore occluded at L9’.

Nevertheless, per the reviewers’ request, we have replaced the term “resting state” to “apo state” throughout the manuscript and leave it to readers to judge whether the apo state represents the resting state.

6. The Fo-Fc map shown for the ligand is in a color that makes it very hard to see, and is too small a part of the figure, and does not show how the ligand is modeled into it. Please make a panel where the full panel is occupied by this map, and shows the ligand in the map, to demonstrate support confidence in modeling.

As requested by the reviewer, an extra-large figure showing the *Fo-Fc* map is now added to the supporting materials as **Fig. S6a**. The alphaxalone fitting to the *Fo-2Fc* electron density is demonstrated in **Fig. 4b** as presented in the previous version.

Reviewer #2 (Remarks to the Author):

The authors have addressed my concerns adequately. I do not have any further comments.

Reviewer #3 (Remarks to the Author):

In the revised manuscript, Chen et al introduced several changes and addenda in response to my criticism. Unfortunately, some of these additions do not do a good job as the manuscript is still permeated of unjustified and confusing statements that should be seriously revised or retracted. Despite I am convinced that the main result of the paper is worth publishing, I expect the authors to take my comments more seriously this time.

In our previous response to reviewers, we did take the reviewers’ comments seriously and addressed all of the comments either with new experimental data or with newly analyzed results. In addition to the heavily-revised main text and new figures suggested by the reviewers, at least four pages out of the 10-page response were dedicated to address Rev#3’s comments. We hope one would see how sincerely we have treated Rev#3’s suggestions and comments.

1. Much confusion arises from the annotation of “resting state” to a structure that presents a closed ion pore and was solved in the absence of ligands (apo), which is NOT sufficient for a non-ambiguous functional assignment as acknowledged by the authors in the Discussion. Therefore, to remove the ambiguity I request all statements below to be modified accordingly.

For ligand-gated ion channels, an apo state without leaking currents is a resting state by definition. Our electrophysiology data (Fig. 1) support the assignment of the resting state – the apo channel has no leaking current, can be activated by agonists, and can also be activated and potentiated by alphaxalone.

There is no convincing experimental evidence (no published apo resting-state structure for GABA_ARs at this point) or theoretical basis to support the widely held opinion that the

resting-state pLGICs must uniformly adopt the same closed-channel conformation with the pore occluded at L9'.

Nevertheless, per the reviewer's request, we have replaced "resting state" with "apo state" in the manuscript.

- Page 2 (abstract) : "Here, we report crystal structures of a new α 1GABAAR chimera (ELIC- α 1GABAAR) in the resting and desensitized states."

- Page 2 (abstract) : "The resting-state structure reveals an unconventional activation gate at the intracellular end of the pore"

- Page 3 (end of introduction) : "Here, we report crystal structures of a new α 1GABAAR chimera in the resting state."

- Page 4 (results) : "We determined x-ray structures of the α 1GABAAR chimera in the resting and desensitized states"

- Page 7 (discussion) : "We have determined both the resting and desensitized structures for ELIC-GABA chimera"

The term "resting" has been replaced with "apo" throughout.

2. There is no evidence in the manuscript supporting the "bottom-up" mechanism for channel activation and desensitization by alphaxalone. The authors simply assume an induced-fit model with alphaxalone binding being the initiating event. Since there is no evidence for it and no crystal structure of any endpoint can provide information on the transition pathway or the sequence of structural events, I request the "bottom-up" keyword to be amended from the title and throughout the Text.

As suggested, we have removed the term "bottom-up" in the manuscript.

3. The comparison of the quaternary structure of the apo chimera versus the alphaxalone-bound state (Fig S5) is enlightening and indicates that, despite the claims on page 5, there is essentially no change in the global receptor twisting or blooming. Both twisting and blooming angles in the two chimera structures should be quantified and reported (see Martin et al. PLoS Comp Biol 2017 for an operational definition) and compared to the functional changes observed in the X-ray structures (Hibbs & Gouaux 2012 and Althoff et al. 2014) and simulations (Martin et al. PLoS Comp Biol 2017) of GluCl. Based on the results, the discussion on page 5 need to be corrected. Also, the review of Cecchini and Changeux (Neuropharmacology 2015) should be used as a reference for the functional implications of twisting/blooming on gating.

As stated on page 5 in the previous revision, one should not expect to see large twisting or blooming because both the apo and alphaxalone-bound structures are closed channels. In the previous revision, we also stated explicitly that we did not see blooming and we saw only small twisting when two structures were aligned (Fig. S5). Quantitative values showing twist angles are now labeled in Fig. S5. Our previous conclusion does not change in this revision.

We have a number of publications^{1,2,3,4}, in which channel twisting and blooming motion are characterized using the conventional lateral tilting angle (δ) and radial tilting angle (θ)^{5,6}. The same method was used in this manuscript. We thank the reviewer for bringing a new simulation paper (Martin et al. PLoS Comp Biol 2017) to our attention that will be useful for future projects. The review article of Cecchini and Changeux (Neuropharmacology 2015) is now cited.

4. Given the strong residue conservation in the -2' to 13' region between GABA and GluCl, the argument that differences in the hydrophobicity pattern in the pore-lining residues produce an upward shift of the constriction point (i.e. to position 2') in the apo chimera (page 8) is not very convincing. Without further arguing, I request the author to include the anionic GlyR $\alpha 1$ and GlyR $\alpha 3$ in the sequence alignment in Fig S11 and leave the reader the opportunity to judge the quality of their argument.

Both $\alpha 1$ and $\alpha 3$ GlyRs are now included in Fig. S11.

REFERENCES

1. Pan J, *et al.* Structure of the pentameric ligand-gated ion channel ELIC cocrystallized with its competitive antagonist acetylcholine. *Nat Commun* **3**, 714 (2012).
2. Mowrey D, *et al.* Asymmetric ligand binding facilitates conformational transitions in pentameric ligand-gated ion channels. *J Am Chem Soc* **135**, 2172-2180 (2013).
3. Willenbring D, Liu LT, Mowrey D, Xu Y, Tang P. Isoflurane alters the structure and dynamics of GLIC. *Biophys J* **101**, 1905-1912 (2011).
4. Tang P, Mandal PK, Xu Y. NMR structures of the second transmembrane domain of the human glycine receptor alpha(1) subunit: model of pore architecture and channel gating. *Biophys J* **83**, 252-262 (2002).
5. Nury H, *et al.* One-microsecond molecular dynamics simulation of channel gating in a nicotinic receptor homologue. *Proc Natl Acad Sci U S A* **107**, 6275-6280 (2010).
6. Althoff T, Hibbs RE, Banerjee S, Gouaux E. X-ray structures of GluCl in apo states reveal a gating mechanism of Cys-loop receptors. *Nature* **512**, 333-337 (2014).

Reviewers' comments:

Reviewer #1 (Remarks to the Author):

I am sorry I was not more clear about what I hoped to see in a Fo-Fc map figure. I would like a panel that illustrates how the authors were guided by the density in modeling of the ligand. For this panel, fine to be in supplementary, please show: Fo-Fc density at a single interface with the ligand present- the goal is to show how well the ligand does/does not fit the density. Thank you.

Reviewer #3 (Remarks to the Author):

The points raised in the previous round of review have been only partially addressed.

On point 1 (resting state annotation):

The caption of Figure 3 still retains a confusing statement. Therefore, I request the following change:

Line 550: CHANGE "chimera in the resting (orange)" WITH "chimera in the apo (orange)"

On point 2 (bottom-up mechanism):

Unlike stated in the response letter, the keyword "bottom-up" was not completely removed from the manuscript. This keyword appears at least four times in the last paragraph of the Discussion and it has been rephrased as "Started from the bottom" in the title. Therefore, I request the removal of "Started from the bottom" from the title and the following changes:

Line 86: CHANGE "starting from" WITH "at"

Line 327: remove "bottom-up"

Line 331: remove "bottom-up"

Line 334: remove "bottom-up"

Line 339: CHANGE "can generate" WITH "is consistent with"

Line 342: remove "bottom-up"

On point 3 (twisting/blooming):

I disagree with the authors' statement that "one should not expect to see large twisting or blooming because both the apo and alphaalone-bound structures are closed channels". Indeed as shown for GlyR (Hibbs & Gouaux 2012 and Althoff et al. 2014) and GluCl (Calimet et al PNAS 2013 and Martin et al PLoS Comp Biol), the resting state is globally more twisted (~10 deg) than the active or desensitized states, such that these collective variables are useful for a functional annotation. Therefore, absolute values for both global twisting and blooming angles should be measured and compared with previous structures and simulations of anionic ligand-gated ion channels (GluCl and GlyR) perhaps in a Table in SI. To this aim, the authors should use the structural definition of the core given in Calimet et al PNAS 2013 and extend it to GABA.

Before all points above are addressed in full, I am unable to recommend this manuscript for publication.

Reviewers' comments:

Reviewer #1 (Remarks to the Author):

I am sorry I was not more clear about what I hoped to see in a Fo-Fc map figure. I would like a panel that illustrates how the authors were guided by the density in modeling of the ligand. For this panel, fine to be in supplementary, please show: Fo-Fc density at a single interface with the ligand present- the goal is to show how well the ligand does/does not fit the density. Thank you.

Per the reviewer's request to facilitate visualization, we show alphaxalone molecules in the Fo-Fc maps in the new Fig. S6a.

Reviewer #3 (Remarks to the Author):

The points raised in the previous round of review have been only partially addressed.

On point 1 (resting state annotation):

The caption of Figure 3 still retains a confusing statement. Therefore, I request the following change:

Line 550: CHANGE "chimera in the resting (orange)" WITH "chimera in the apo (orange)" "Resting" is now replaced with "apo".

On point 2 (bottom-up mechanism):

Unlike stated in the response letter, the keyword "bottom-up" was not completely removed from the manuscript. This keyword appears at least four times in the last paragraph of the Discussion and it has been rephrased as "Started from the bottom" in the title. Therefore, I request the removal of "Started from the bottom" from the title and the following changes:

We overlooked these residual "bottom-up" phrases in the Discussion and have now removed them.

As we reasoned in our previous response to reviewers and the main text, our results from functional/mutagenesis measurements (Fig. 4) and x-ray structures (Fig. 2) show unambiguously that the channel is activated purely by alphaxalone bound to the bottom of the TMD. We need to reflect these findings in the title and text. We have taken the reviewer's suggestion and replaced the word "from" to "at" in the title and text.

Line 86: CHANGE "starting from" WITH "at" We took the reviewer's suggestion and replaced "starting from" with "starting at".

Line 327: remove "bottom-up" Removed

Line 331: remove "bottom-up" Removed

Line 334: remove "bottom-up" Removed

Line 339: CHANGE "can generate" WITH "is consistent with" this has been changed to "can introduce"

Line 342: remove "bottom-up" Removed

On point 3 (twisting/blooming):

I disagree with the authors' statement that "one should not expect to see large twisting or

blooming because both the apo and alphaxalone-bound structures are closed channels”. Indeed as shown for GlyR (Hibbs & Gouaux 2012 and Althoff et al. 2014) and GluCl (Calimet et al PNAS 2013 and Martin et al PLoS Comp Biol), the resting state is globally more twisted (~10 deg) than the active or desensitized states, such that these collective variables are useful for a functional annotation. Therefore, absolute values for both global twisting and blooming angles should be measured and compared with previous structures and simulations of anionic ligand-gated ion channels (GluCl and GlyR) perhaps in a Table in SI. To this aim, the authors should use the structural definition of the core given in Calimet et al PNAS 2013 and extend it to GABA.

As we stated in the manuscript and in the previous response to the reviewers, there is no resting state structure reported for GlyR.

The paper “Hibbs & Gouaux 2012” mentioned by the reviewer does not exist. We assume it was simply a typo and the reviewer meant “Hibbs & Gouaux 2011.” However, both papers (Hibbs & Gouaux, 2011 and Althoff et al. 2014) are related to GluCl, not GlyR.

Note that the closed/resting state GluCl (also called apo GluCl) structure is actually a structure of the GluCl-Fab complex (Althoff et al. 2014). As shown in Supplementary Figure 12b in the paper by Hibbs & Gouaux, 2011, Fab significantly reduces the binding of the agonist glutamate to the extracellular domain of GluCl. Unfortunately, the presence of Fab is required for obtaining high quality GluCl crystals. However, it is uncertain how much the Fab binding affects the GluCl structure, especially after removing it in the MD simulations.

The authors of Hibbs & Gouaux 2011 have proposed that the ivermectin-bound GluCl represents a potentially open/activated state. Thus, a comparison of the closed vs. open GluCl in the simulations shows larger conformational changes, including larger blooming and twisting. In our case, however, relative to the aligned structure of the apo ELIC- α 1GABAAR, the alphaxalone-bound structure shows only a small counterclockwise twist (0.80°) of the ECD and a clockwise twist (1.37°) of the TMD around the pore axis (**Fig. S5**). These quantitative values were reported in our previous revision. We remain the same conclusion that there is no significant blooming between our apo and alphaxalone-bound structures. Per the reviewer’s request, we have included quantitative values in the Results section to show the insignificant blooming change: Relative to the aligned structure of the apo ELIC- α 1GABAAR, the alphaxalone-bound structure shows 1.3° inward and 0.18° outward radial tilt from the pore axes in the ECD and TMD, respectively.

Prolonged MD simulations as reported in *Calimet et al PNAS 2013 and Martin et al PLoS Comp Biol* can generate an ensemble of structures showing larger twisting or blooming between different functional states. We may take the same approach in our future work; however, the current manuscript largely focuses on experimental structures and functions.